# Exact instanton transseries for quantum mechanics

**Alexander van Spaendonck⋆ and Marcel Vonk†**

Institute of Physics, University of Amsterdam, Science Park 904,
1090 GL Amsterdam, The Netherlands

⋆ a.b.n.vanspaendonck@uva.nl , † m.l.vonk@uva.nl

## Abstract

We calculate the instanton corrections to energy spectra of one-dimensional quantum mechanical oscillators to all orders and unify them in a closed form transseries description. Using alien calculus, we clarify the resurgent structure of these transseries and demonstrate two approaches in which the Stokes constants can be derived. As a result, we formulate a minimal one-parameter transseries for the natural nonperturbative extension to the perturbative energy, which captures the Stokes phenomenon in a single stroke. We derive these results in three models: quantum oscillators with cubic, symmetric double well and cosine potentials. In the latter two examples, we find that the resulting full transseries for the energy has a more convoluted structure that we can factorise in terms of a minimal and a median transseries. For the cosine potential we briefly discuss this more complicated transseries structure in conjunction with topology and the concept of the resurgence triangle.

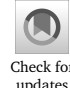 Check for updates

# 1   Introduction

It has been tested and verified in several quantum physical settings, that perturbative series encode information about many – but not necessarily all – nonperturbative effects. This information can be extracted from the nonperturbative ambiguities that arise as we attempt to resum the generally divergent perturbative series into a well-defined and smooth function.

The mathematical tool to extract nonperturbative information from asymptotically divergent series is resurgence [1]. In the realm of ODE's, where the theory of resurgence was conceived, there is a very clear understanding of why these ambiguities arise and how to deal with them. There, resurgence tells us that both perturbative and nonperturbative asympotics should be unified in a single object called a *transseries*. These transseries and their resurgent structure, expressed in the language of *alien calculus*, illustrate how all ambiguities arising from the resummation procedure are eventually cancelled, leading to a well-defined answer where the only remaining ambiguities are related to boundary conditions of the ODE.

The nonperturbative treatment of one-dimensional quantum mechanical oscillators suggests [2] that for these, the perturbative energy spectrum together with all its instanton corrections[1] should also be unified in a single transseries expression. The modern approach to calculate these instanton corrections is through the formulation of *exact quantisation conditions*, which were derived by Zinn-Justin [3] using instanton calculus and by Voros [4,5] and subsequently Delabaere, Dillinger and Pham [6] using exact WKB techniques. Another approach for obtaining these exact quantisation conditions is the uniform WKB method [–10]. As is the case for ODEs, nonperturbative effects in quantum mechanical problems are often encoded in the *large order behaviour* of the perturbative coefficients. The large order behaviour of the perturbative coefficients in quantum mechanics models was initially studied by Bender and Wu in [11], and subsequent works have explored its intricate relation to instantons and nonperturbative ambiguities, e.g. [12–15] and have employed various tools from resurgence to study these nonperturbative aspects even further, e.g. [16,17].

In this work our goal will be to go beyond the status quo and elucidate the structure of transseries in quantum mechanics. We will follow the approach of Álvarez and Casares [,10] for obtaining the instanton corrections to the energy of the quantum states. We clarify some details of this method and extend it to all nonperturbative orders in closed form, formulating a complete transseries expression for the energy of the states. We will then advocate that the natural transseries extension to the perturbative energy $E$ takes the form of the following

---

[1]A remark on terminology: in this paper, following the conventions in the resurgence literature, we use the term 'instanton' for exact instanton solutions as well as for nonperturbative effects related to quasi-instantons. Similarly, we will describe both of these effects as 'saddle point contributions', even though strictly speaking quasi-instantons do not correspond to exact saddle points.

generic *one-parameter transseries*:

$$\hat{E}(t,\sigma;\hbar) = E + \hbar \sum_{n=1}^{\infty} \sum_{m=0}^{n-1} u_{n,m} \sigma^{m+1} \left(\hbar \frac{\partial}{\partial t}\right)^m \left(\frac{\partial E}{\partial t} e^{-\frac{n}{\hbar}t_D}\right). \tag{1}$$

The details of this equation will be clarified in the core of the paper, but the most important thing to note is that this equation expresses the energy transseries $\hat{E}$ in terms of purely *perturbative* quantities $E(t;\hbar)$ and $t_D(t;\hbar)$, a transseries parameter $\sigma$, and some model specific coefficients $u_{n,m}$ for which we will derive a closed form expression. We will demonstrate in detail how alien calculus encodes the Stokes phenomenon of (1) and will derive its Stokes constants in two different ways.

Our approach is strongly influenced by a recent line of work investigating the close connection between quantum mechanical oscillators and topological strings. It has been argued in [18] that *quantum periods*, which are the essential building blocks of the aforementioned quantisation conditions, are governed by the holomorphic anomaly equations [19] of the refined topological string in the Nekrasov-Shatashvili background [20]. Subsequently, building on ideas from [21, 22], the nonperturbative solution to these holomorphic anomaly equations was studied in [23] and finally solved for in exact form in [24] – also in the so-called selfdual background [25]. This nonperturbative solution, that appears in the form of a transseries for the *quantum prepotential* or Nekrasov-Shatashvili free energy, relates directly to the ordinary energy $E$ through a relation known as the *PNP relation*. We will exploit this relation, allowing us to compare to, and utilise the results of [24] for our goals.

To develop and illustrate our ideas, we will consider three models: quantum mechanics in a cubic, a symmetric double well and a cosine potential. Each of these will add an extra layer of complexity to the story and will help us get a deeper understanding of the role that transeries play in quantum mechanics. The cubic will act as a warm-up example to introduce our techniques and illustrate the core notions of our work that lead to (1), an object that we call the minimal resurgent transseries. The double well potential has a more elaborate transseries structure that factorises (in a way we will make precise) in terms of the minimal transseries and another object that we call the median transseries. Finally, we discuss the cosine potential where the energy levels at the nonperturbative level split into bands. Each band consists of a continuous spectrum of energies that are characterised by an angle $\theta$. The dependence of the instantons on $\theta$ then allows us to identify topological sectors in the model and explain how our transseries structure relates to the concept of a 'resurgence triangle' as described in [26, 27].

This paper is organised as follows. In section 2 we introduce the most important concepts and tools of resurgence, alien calculus and the all-orders WKB method. In section 3 we then start with a study of the cubic oscillator: we introduce the closed form expression for the nonperturbative energy spectrum, illustrate two ways in which the Stokes phenomenon of the transseries can be decoded, including a computation of the exact values for the Stokes constants, and arrive at the generic one-parameter transseries structure (1). In section 4 we then move to the double well potential and use the same techniques to derive its energy transseries. We clarify its resurgent structure and illustrate how it factorises in terms of a minimal and a median transseries. In section 5 we then adress the cosine potential, for which we repeat the procedure for obtaining the full energy transseries and comment on its topological structure in comparison with the resurgence triangle. Finally, in section 6 we state our conclusions, and three appendices are included in which we derive the PNP relation (A), discuss the Stokes phenomenon of the quantum prepotential (B) and elaborate more on median resummation and the median transseries (C).

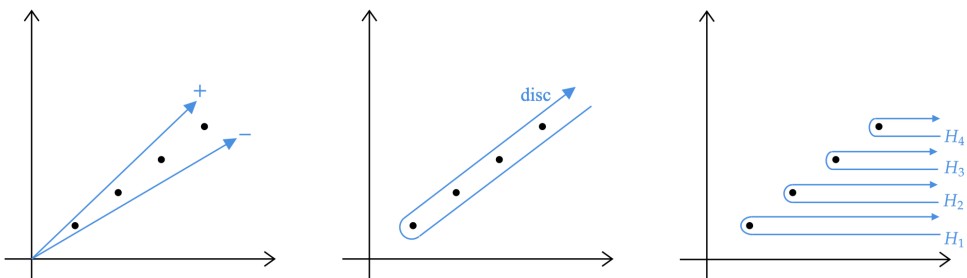

Figure 1: The Borel plane of $\mathcal{B}\varphi$: on the left both $\mathcal{S}_{\theta^+}\varphi$ and $\mathcal{S}_{\theta^-}\varphi$, in the middle their difference $\mathrm{disc}_\theta \varphi(x)$ and on the right also the discontinuity, but now decomposed in Hankel contours $H_\omega$.

## 2 Preliminaries

In this section we introduce the most important tools and concepts that are necessary for the calculations that will follow in subsequent sections. This section is subdivided in two subsections discussing some basics of resurgence and alien calculus and subsequently the all-orders WKB method and the transseries solution to the holomorphic anomaly equation.

### 2.1 Resurgence and alien calculus

Let us recall the aspects of resurgence (originally introduced in Écalle's seminal work [1]) that are most relevant to us, and mention some essential tools and notation that will be important in the forthcoming calculations. For a thorough introduction to the topic of resurgence from a physics point of view we recommend reading [28–30], and for the more mathematically inclined reader we recommend [31, 32].

We start by considering a formal power series $\varphi(x) = \sum_{n=0}^\infty a_n x^n$ whose coefficients $a_n$ diverge factorially: $a_n \sim n!$. Its *Borel transform* is defined as

$$\mathcal{B}\varphi(s) = \sum_{n=0}^\infty \frac{a_{n+1}}{n!} s^n \,. \tag{2}$$

It allows us to resum the asymptotic series $\varphi(x)$ – minus its constant term, which we can add again at the end of the procedure – using the *Borel resummation* $\mathcal{S}_\theta$:

$$\mathcal{S}_\theta \varphi = \int_0^{e^{i\theta}\infty} \mathcal{B}\varphi(s)\, e^{-s/x}\, ds \,, \tag{3}$$

where the integration contour runs along a straight line from zero to infinity under an angle $\theta$ in the complex $s$-plane, which we will henceforth call the *Borel plane*. Here, one assumes that the formal series (2) can be summed and analytically continued into a function of $s$, so that the integral (3) makes sense. In general, the function $\mathcal{B}\varphi(s)$ will have a number of isolated singularities in the Borel plane. Whenever one or more of these singularities lie on the integration contour, we call this ray a *Stokes line* and we find that our resummation procedure is ill-defined. The resolution to this problem is to slightly deform the contour either above ($\mathcal{S}_{\theta^+}$) or below the singularities ($\mathcal{S}_{\theta^-}$), leading to two distinct resummations, as depicted in the left graph of figure 1. The difference between these *lateral Borel summations* constitutes the *discontinuity*:

$$\mathrm{disc}_\theta \varphi(x) = \mathcal{S}_{\theta^+}\varphi(x) - \mathcal{S}_{\theta^-}\varphi(x) \,. \tag{4}$$

The difference between these two quantities can be computed by integrating around all the singularities that lie on the ray with angle $\theta$, as shown in figure 1 in the middle. Let $\Omega_\theta$ be the collection of indices[2] that label the singularities that lie along such a ray in the Borel plane of $\mathcal{B}[\varphi](s)$. That is, for every $\omega \in \Omega_\theta$, there is a singularity located at $\mathcal{A}_\omega$ with $\arg(\mathcal{A}_\omega) = \theta$. The contour above can then be decomposed in terms of multiple Hankel contours, each of them 'grabbing' a single singularity labeled by one or more $\omega \in \Omega_\theta$, as shown in figure 1 on the right. For many resurgent functions, it turns out that the Borel transform (2), expanded near each of the singularities $\mathcal{A}_\omega$, behaves as

$$\mathcal{B}\varphi(s) \simeq \frac{S_\omega a_{\omega,0}}{2\pi i(s - \mathcal{A}_\omega)} + \frac{S_\omega \log(s - \mathcal{A}_\omega)}{2\pi i} \mathcal{B}[\varphi_\omega](s - \mathcal{A}_\omega) + \text{regular}. \tag{5}$$

Here $a_{\omega,0}$ and $S_\omega$ are constants and $\mathcal{B}\varphi_\omega(s)$ is an analytic function. We can then add up the contributions from the different Hankel contours to calculate the discontinuity, and obtain

$$\text{disc}_\theta \, \varphi(x) = \sum_{\omega \in \Omega_\theta} S_\omega e^{-\mathcal{A}_\omega/x} \mathcal{S}_{\theta^-} \varphi_\omega(x). \tag{6}$$

The $\varphi_\omega(x) = \sum_{n=0}^\infty a_{\omega,n} x^n$ – where the constant term comes from the leading term in (5) – are then new diverging formal power series that have 'resurged' from the Borel plane, and the $S_\omega$ are known as *Stokes constants*.[3] Note that the above equation concerns resummed power series, that is: Functions. The same phenomenon can also be expressed at the level of asymptotic expansions, by introducing the *Stokes automorphism*, which is defined by the relation

$$\mathcal{S}_{\theta^+} = \mathcal{S}_{\theta^-} \circ \mathfrak{S}_\theta. \tag{7}$$

It essentially tells us what the correct way is to analytically continue an asymptotic expansion across a Stokes line. Using this definition, we can rewrite equation (6) in terms of asymptotics:

$$\mathfrak{S}_\theta \varphi(x) = \varphi(x) + \sum_{\omega \in \Omega_\theta} S_\omega \varphi_\omega(x) e^{-\mathcal{A}_\omega/x}. \tag{8}$$

The appearance of the additional nonperturbative contributions on the right hand side, encoded in the *transmonomials* $e^{-\mathcal{A}_\omega/x}$, is called the *Stokes phenomenon*. The above discussion suggests that to study $\mathfrak{S}_\theta$, we should extend the notion of an asymptotic expansion from a power series $\varphi(x)$ to a *transseries*:

$$\hat{\varphi}(\vec{\sigma}, x) = \varphi(x) + \sum_{\omega \in \Omega_\theta} \sigma_\omega \varphi_\omega(x) e^{-\mathcal{A}_\omega/x}. \tag{9}$$

A remark regarding notation: in this paper asymptotic quantities *without* a hat are ordinary power series whereas those *with* a hat are transseries. The $\sigma_\omega$ in the above expression are called *transseries parameters* and we have collected them on the left hand side in a single vector $\vec{\sigma}$ for convenience.[4] By introducing these parameters we have constructed a *family* of

---

[2]We choose $\Omega_\theta$ to be a set of indices rather than a set of singularities since, as we will see in more detail later, several different indices $\omega$ may correspond to the same singularity location. This will imply that a sector with a certain nonperturbative 'weight' $e^{-\mathcal{A}_\omega/x}$ actually gets multiplied by a sum of several different asymptotic series.

[3]In some of the literature, these constants are called *Borel residues*. The Stokes constants are then a set of equivalent constants that determine the discontinuity, usually defined by the *bridge equations* of resurgence. In many physical problems however, we do not have such a bridge equation and we call the constants $S_\omega$ Stokes constants instead.

[4]Often, it is not necessary to introduce an independent transseries parameter $\sigma_\omega$ for *every* $\omega \in \Omega_\theta$. For example, in solutions to nonlinear ODEs, one often has that $\sigma_{nA} = (\sigma_A)^n$. For now, we will work with the generic case where all transseries parameters are independent, but in our examples we will generally only see a single independent transseries parameter $\sigma$ appear.

transseries expressions, and we then interpret the Stokes automorphism as an operation acting on this family:

$$\mathfrak{S}_0 \,\hat{\varphi}(\vec{\sigma}, x) = \hat{\varphi}(g(\vec{\sigma}), x). \tag{10}$$

This provides us with a qualitative picture of the Stokes phenomenon. For a more quantitative understanding, we need to deduce the exact mapping $g(\vec{\sigma})$ on the transseries parameter space. This is in general a complicated problem, which we studied earlier for e.g. the Painlevé I ODE in [33]. In order to achieve this in the present setting, we need to introduce the notion of alien calculus [1], which introduces a new set of derivative operators $\Delta_{\mathcal{A}_\omega}$ called *alien derivatives*. These operators are labeled by the location $\mathcal{A}_\omega$ of the singularities in the Borel plane and act on formal power series producing new formal power series:

$$\Delta_{\mathcal{A}_\omega} : \mathbb{C}[[x]] \to \mathbb{C}[[x]]. \tag{11}$$

These operators turn out to have some special properties. First of all, they satisfy the Leibniz rule when acting on a product of formal power series. Secondly, the Stokes automorphism can be decomposed in terms of these new operators as follows:

$$\mathfrak{S}_\theta = \exp\left(\sum_\omega e^{-\mathcal{A}_\omega/x} \Delta_{\mathcal{A}_\omega}\right). \tag{12}$$

For the third property, it is convenient to introduce the *pointed alien derivative*

$$\dot{\Delta}_{\mathcal{A}_\omega} \equiv e^{-\mathcal{A}_\omega/x} \Delta_{\mathcal{A}_\omega}. \tag{13}$$

This operator has the very useful property that it commutes with differentiation with respect to the expansion variable $x$:

$$\left[\dot{\Delta}_{\mathcal{A}_\omega}, \frac{\partial}{\partial x}\right] = 0. \tag{14}$$

With the language of alien calculus, we can now define the *minimal resurgent structure* [34] associated to the original series $\varphi(x)$ and the angle[5] $\theta$:

$$\mathfrak{B}_{(\varphi,\theta)} = \{\, \varphi_\omega(x) \,|\, \omega \in \Omega_\theta \,\}. \tag{15}$$

Loosely speaking, it contains all the formal power series that resurge from the series $\varphi(x)$. More precisely, using the alien derivatives we can define $\mathfrak{B}_{(\varphi,\theta)}$ as the smallest set of power series that is closed under the Stokes automorphism $\mathfrak{S}_\theta$ and that includes $\varphi(x)$. That is, for $\arg(\mathcal{A}) = \theta$ and $\omega, \omega' \in \Omega_\theta$, we have

$$\Delta_{\mathcal{A}} \varphi_\omega = \sum_{\omega'} c^{\mathcal{A}}_{\omega\omega'} \varphi_{\omega'}, \tag{16}$$

where the $c^{\mathcal{A}}_{\omega\omega'}$ are some constants that we can express in terms of the Stokes constants. In the case where all the formal power series $\varphi_\omega$ in (9) reside in the minimal resurgent structure (15) – therefore making the transseries itself 'closed' – we call the corresponding transseries a *minimal resurgent transseries*. In the examples that we study in this paper, we will encounter such minimal transseries, but also transseries which contain additional 'disconnected' power series which make the transseries no longer minimal.

Lastly, in this paper we will mostly be concerned with formal power series and transseries whose coefficients depend on some other parameter, say $y$ – e.g.

$$\varphi(y; x) = \sum_{n=0}^{\infty} a_n(y) x^n. \tag{17}$$

---

[5]We add the angle to the definition so that we can focus on a single Stokes transition.

The study of such objects goes under the name of *parametric resurgence*. When we now substitute $y$ itself with a formal power series $\phi(x) = \sum_{n=0}^{\infty} b_n x^n$, we obtain a completely new power series[6] $\chi(x) = \varphi(\phi(x); x)$ whose resurgence properties might seem unknown. However, in terms of alien calculus, the substitution can be understood very well and for the pointed alien derivatives one has the following chain rule of [17] which reads

$$\dot{\Delta}_{\mathcal{A}} \chi(x) = \left(\dot{\Delta}_{\mathcal{A}} \varphi\right)(y; x)\Big|_{y=\phi(x)} + \frac{\partial \varphi}{\partial y}(\phi(x); x)\, \dot{\Delta}_{\mathcal{A}} \phi(x). \tag{18}$$

One can think of this rule as the 'alien analogue' of the chain rule in ordinary calculus.

## 2.2 All-orders WKB and the quantum prepotential

We will be interested in one-dimensional quantum mechanics problems described by the time-independent Schrödinger equation

$$-\frac{\hbar^2}{2} \psi''(x) + V(x)\psi(x) = E\, \psi(x). \tag{19}$$

Solutions to this equation are constructed via the *all-orders WKB* Ansatz [35]

$$\psi(x) = \exp\left(\frac{\mathrm{i}}{\hbar} \int^x S(x'; \hbar)\, \mathrm{d}x'\right), \tag{20}$$

where $S(x; \hbar) = \sum_{n=0}^{\infty} p_n(x)\hbar^n$ is a solution to the *Riccati equation*

$$S^2 - \mathrm{i}\hbar \frac{\partial S}{\partial x} = 2(E - V(x)). \tag{21}$$

The solution for the leading piece $p = p_0(x)$ of the Riccati solution $S(x; \hbar)$ defines a Riemann surface $\Sigma$ that we call the *WKB curve*:

$$\Sigma: \; p^2 = 2E - 2V(x). \tag{22}$$

In this paper we will consider the cubic, double well and cosine potentials which effectively have an underlying genus one elliptic WKB curve (see e.g. [36]). Most importantly, we can integrate the Riccati solution $S(x; \hbar)$ along nontrivial cycles $\gamma \in H_1(\Sigma)$ in the WKB curve to obtain *quantum periods*

$$\Pi_\gamma = \oint_\gamma S(x; \hbar)\, \mathrm{d}x. \tag{23}$$

The solution to the Riccati equation can be separated into even and odd parts:

$$S(x; \hbar) = S^{\mathrm{even}}(x; \hbar) + S^{\mathrm{odd}}(x; \hbar), \tag{24}$$

where the even and odd parts contain only even or odd powers of $\hbar$, respectively.[7] One can then show that the odd part is a total derivative and that its contribution to the quantum

---

[6]This substitution does not necessarily produce a well-defined power series. The series $\varphi(y; x)$ therefore needs to satisfy some mild conditions explained [17].

[7]The Ricatti equation has two possible solutions $S^{(\pm)}$ determined by the sign of the initial term $p_0^{(\pm)}(x) = \pm\sqrt{2E - 2V(x)}$. Formally, the even and odd solutions are then defined as

$$S^{\mathrm{even}} = \frac{S^{(+)} + S^{(-)}}{2}, \qquad \text{and} \qquad S^{\mathrm{odd}} = \frac{S^{(+)} - S^{(-)}}{2},$$

which in the absence of quantum corrections to the potential imply the above-mentioned separation in even and odd powers of $\hbar$.

periods vanishes. As a consequence, we can expand (23) asymptotically as

$$\Pi_\gamma(E;\hbar) = \sum_{n=0}^{\infty} \left( \oint_\gamma p_{2n}(x,E)\,\mathrm{d}x \right) \hbar^{2n} = \sum_{n=0}^{\infty} \Pi_{\gamma,n}\hbar^{2n}\,. \tag{25}$$

If we fix the energy $E$ for a moment, one can check that the coefficients of the quantum periods grow double factorially:

$$\Pi_{\gamma,n} \sim (2n)!\,, \tag{26}$$

and hence we need to Borel resum these asymptotic expansions in order to produce sensible functions.

The analytically continued Borel transforms of the quantum periods have singularities in the Borel plane. These singularities organise themselves on rays across which the Stokes automorphism $\mathfrak{S}_\theta$ is given by the *Delabaere-Dillinger-Pham* formula [16, 17] (see also appendix A of [37]). To state this formula, we first need to define the *Voros symbol*

$$\mathcal{V}_\gamma = \mathrm{e}^{\frac{\mathrm{i}}{\hbar}\Pi_\gamma(E;\hbar)}\,. \tag{27}$$

Now, let us consider the case in which for a given angle $\theta = \arg(\hbar)$ there is one[8] cycle $\gamma'$ whose associated quantum period satisfies $\frac{\mathrm{i}}{\hbar}\Pi_{\gamma',0} \in \mathbb{R}_{<0}$. If the cycles $\gamma$ and $\gamma'$ *intersect*, then the Borel transform of the quantum period $\Pi_\gamma$ has a ray with singularities on it along the angle $\theta$ in its Borel plane. The Delabaere-Dillinger-Pham formula now tells us that across that Stokes ray, the Stokes automorphism for the associated Voros symbol $\mathcal{V}_\gamma$ is given by

$$\mathfrak{S}_\theta \mathcal{V}_\gamma = \mathcal{V}_\gamma \left(1 + \mathcal{V}_{\gamma'}\right)^{\langle\gamma',\gamma\rangle}\,, \tag{28}$$

where $\langle\gamma,\gamma'\rangle$ is the intersection number of the two cycles.

Since all the examples in this paper will involve an underlying genus 1 curve, we have a basis of two independent cycles $\gamma_A, \gamma_B \in H_1(\Sigma)$. In general, our potentials will be real, and following standard conventions we denote cycles around minima of the potential, where particles can follow classical trajectories, as A-cycles, while labeling 'tunneling' cycles around maxima as B-cycles. In what follows, we will speak of the *quantum A-period*, which is defined as

$$t(E;\hbar) = \sum_{n=0}^{\infty} t_n(E)\hbar^{2n} = \frac{1}{2\pi}\Pi_{\gamma_A}(E;\hbar)\,. \tag{29}$$

There will also be a *dual period* or *quantum B-period* that we define as

$$t_D(E;\hbar) = \sum_{n=0}^{\infty} t_{D,n}(E)\hbar^{2n} = -\mathrm{i}\Pi_{\gamma_B}(E;\hbar)\,. \tag{30}$$

The numerical prefactors in these definitions are conventional, and chosen to agree with expressions in the literature. As a consequence of these prefactors, in the quantum mechanical problems that we study here, where all turning points $x_i$ defined by $V(x_i) = E$ lie on the real axis, the $t_0$ and $t_{D,0}$ will be real and positive. Figure 2 gives an impression of the respective Borel planes and shows that the quantum A-period $t$ is non-Borel summable when $\hbar$ is real, whereas the dual quantum period is non-Borel summable when $\hbar$ is purely imaginary. As a consequence, the Delabaere-Dillinger-Pham formula tells us how the quantum periods jump

---

[8]In general, there may be multiple cycles $\gamma_i$ satisfying this condition. In that case the DDP formula generalises to

$$\mathfrak{S}_\theta \mathcal{V}_\gamma = \mathcal{V}_\gamma \prod_i \left(1 + \mathcal{V}_{\gamma_i}\right)^{\langle\gamma_i,\gamma\rangle}\,.$$

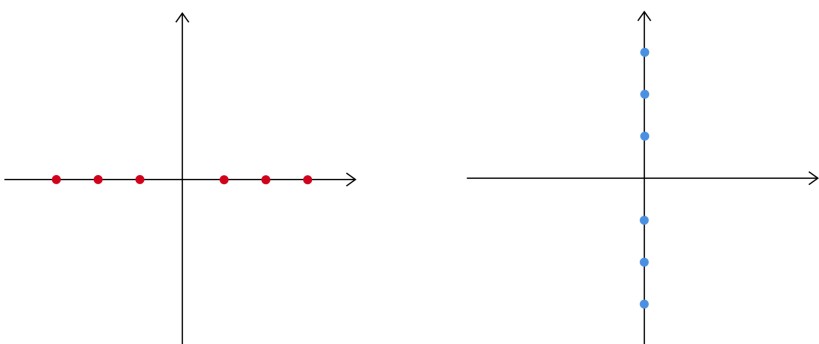

Figure 2: A sketch of the Borel planes of the quantum A-period $t(E;\hbar)$ on the left and the quantum B-period $t_D(E;\hbar)$ on the right. The singularities are evenly spaced and continue indefinitely.

across the four Stokes rays located at $\arg(\hbar) \in \frac{\pi}{2}\mathbb{Z}$. For example, along the rays $\arg(\hbar) = 0$ and $\arg(\hbar) = \frac{\pi}{2}$ the asymptotic expansions of the quantum periods jump in the following way:

$$
\begin{aligned}
\mathfrak{S}_0\, t(E;\hbar) &= t(E;\hbar) + \langle \gamma_B, \gamma_A \rangle \frac{\hbar}{2\pi i} \log\left(1 + e^{-t_D(E;\hbar)}\right), && \text{for } \arg(\hbar) = 0, \\
\mathfrak{S}_{\frac{\pi}{2}}\, t_D(E;\hbar) &= t_D(E;\hbar) + \langle \gamma_A, \gamma_B \rangle \hbar \log\left(1 + e^{-2\pi i t(E;\hbar)}\right), && \text{for } \arg(\hbar) = \frac{\pi}{2}.
\end{aligned}
\tag{31}
$$

In all of our examples we pick orientations of $\gamma_A, \gamma_B$ such that $\langle \gamma_A, \gamma_B \rangle = -\langle \gamma_B, \gamma_A \rangle = 1$.

One can invert the quantum A-period to express the energy $E$ as a formal power series in $\hbar$ with coefficients depending on $t$:

$$
E(t;\hbar) = \sum_{n=0}^{\infty} E_n(t)\hbar^{2n}.
\tag{32}
$$

The coefficient functions $E_n(t)$ are then determined by solving $E(t(E;\hbar);\hbar) = E$. They also grow double factorially, just like the $\Pi_{\gamma,n}$ in (26). In the topological string theory literature, the above series is often called the *quantum mirror map*. There is also the *classical mirror map*, which is given by the leading piece $E_0(t)$ and which is the inverse of the classical period $t_0(E)$.

The dual period $t_D(E;\hbar)$ also defines a *quantum prepotential* or *Nekrasov-Shatashvili free energy*[9] [20]

$$
F(t;\hbar) = \sum_{n=0}^{\infty} F_n(t)\hbar^{2n}.
\tag{33}
$$

For our genus 1 examples, up to an integration constant this quantity is defined by the equation

$$
t_D(t;\hbar) = \frac{\partial}{\partial t} F(t;\hbar).
\tag{34}
$$

Note that on the left hand side of this equation we abuse the notation slightly: the original dual quantum period $t_D$ defined in (30) is a function of $(E, \hbar)$ which relates to $t_D(t;\hbar)$ shown here in the following way:

$$
t_D(t;\hbar) \equiv t_D(E(t;\hbar);\hbar).
\tag{35}
$$

---

[9]In this work we will mostly use the term *quantum prepotential*, for two good reasons: First of all we are essentially studying the quantisation of elliptic curves and our discussion will be quite remote from supersymmetric gauge theory context that the story connects to. Secondly, when we discuss the PNP relation in upcoming sections it might be confusing to speak of both of the energy $E$ and the NS free energy $F$.

On the right hand side of this expression we see the original definition (30) of the quantum dual period with the argument $E$ substituted by the quantum mirror map $E(t;\hbar)$, therefore making it a composition of series in $\hbar$. If we switch back to the independent variables $(E,\hbar)$, then equation (34) becomes

$$t_D(E;\hbar) = \frac{\partial}{\partial t} F(t(E;\hbar);\hbar). \tag{36}$$

Note moreover that the definition of the quantum prepotential involves a choice of *frame*: we can also exchange $t$ and $t_D$ in (34) leading to a different quantum prepotential $F_D$ defined by

$$t(t_D;\hbar) = \frac{\partial}{\partial t_D} F_D(t_D;\hbar), \tag{37}$$

where now $(t_D,\hbar)$ are the independent variables. In fact, one can choose two arbitrary $\mathbb{Z}$-linear combinations of cycles as new A- and B-cycles, as long as they form a symplectic basis for the space of all cycles, and define an associated quantum prepotential – but all of the quantum prepotentials one obtains in this way can in the end be related to one another using an appropriate Legendre transformation.

The quantum prepotential turns out to be a factorially divergent power series and is therefore a suitable object to study using resurgence. Let us introduce – again by a slight abuse of notation – the quantum prepotential $F(E;\hbar)$, which relates to our quantum prepotential $F(t;\hbar)$ in (33) as

$$F(E;\hbar) = F(t_0(E);\hbar). \tag{38}$$

The transseries extension to this series and its resurgent structure were studied in [24], and here we briefly recall some key aspects that will be necessary for what is to come.[10] The Borel transform of the quantum prepotential $F(E;\hbar)$ will have rays along which evenly spaced singularities lie. Let us take such a ray where the singularities lie at $n\mathcal{A}$ with $n \in \mathbb{Z}_{>0}$. Then a transseries extension of $F$ which includes all the associated nonperturbative sectors will have the general form

$$\hat{F} = \sum_{n=0}^{\infty} F^{(n)}, \qquad \text{where} \qquad F^{(n)} \sim e^{-n\mathcal{A}/\hbar}. \tag{39}$$

Here, the perturbative quantum prepotential $F$ is encoded in $F^{(0)} = F - F_0$, where following standard conventions the leading term is subtracted to obtain nicer-looking holomorphic anomaly equations. The first three nonperturbative sectors [24] have the form

$$\begin{aligned}
F^{(1)} &= \tau_1 e^{-\mathcal{G}/\hbar}, \\
F^{(2)} &= \left(-\frac{\tau_2}{4} + \tau_1^2 \frac{D\mathcal{G}}{2\hbar^2}\right) e^{-2\mathcal{G}/\hbar}, \\
F^{(3)} &= \left(\frac{\tau_3}{9} - \tau_1\tau_2 \frac{D\mathcal{G}}{2\hbar^2} + \tau_1^3 \left(\frac{(D\mathcal{G})^2}{2\hbar^4} - \frac{D^2\mathcal{G}}{6\hbar^3}\right)\right) e^{-3\mathcal{G}/\hbar}.
\end{aligned} \tag{40}$$

Here, the $\tau_k$ are yet undetermined constants that we will interpret as transseries parameters. The operator $D$ and the quantity $\mathcal{G}$ are quite subtle objects that depend on which Stokes line we are studying. For a complete and thorough explanation we refer to [23–25], but for our purposes it suffices to know that $\mathcal{G}$ has the generic form

$$\mathcal{G} = \mathcal{A} + DF^{(0)}, \quad \text{with} \quad \mathcal{A} = \alpha \frac{\partial F_0}{\partial t} + \beta t_0 + \gamma, \tag{41}$$

---

[10]The key result that we are after is the action of the alien derivatives on the perturbative quantum prepotential $F(E;\hbar)$, shown in (46). For the purpose of understanding the main results of the present paper, the finer details of the quantum prepotential transseries $\hat{F}$ presented here can be skipped upon first reading.

where $\alpha$, $\beta$ and $\gamma$ are constants that express the location $\mathcal{A}$ of the first singularity in the Borel plane in terms of the classical A- and B-periods. In our 1d quantum mechanical models, there will only be two relevant frames to consider: the first, and for us most important frame is relevant for the Stokes transition along the real $\hbar$-axis. It gives rise to nonperturbative corrections to the quantum prepotential with

$$\mathcal{G} = \alpha \frac{\partial F}{\partial t}, \quad \text{and} \quad \mathrm{D} = \alpha \frac{\partial}{\partial t}. \tag{42}$$

Secondly, as we mentioned there is a Stokes phenomenon across the imaginary $\hbar$-axis giving rise to nonperturbative corrections to the quantum prepotential with

$$\mathcal{G} = \beta t_0, \quad \text{and} \quad \mathrm{D} = 0. \tag{43}$$

In the examples that were studied numerically in [24], the Stokes constants $S_\omega$ (see (5)) scale with $\hbar^2$. To remove this scaling we will slightly modify their normalisation by shifting $\tau_k \to \hbar^2 \tau_k$. In this new normalisation, the first three transseries sectors therefore take the form

$$\hbar^{-2} F^{(1)} = \tau_1 e^{-\mathcal{G}/\hbar},$$
$$\hbar^{-2} F^{(2)} = \left( -\frac{\tau_2}{4} + \frac{1}{2} \tau_1^2 \mathrm{D}\mathcal{G} \right) e^{-2\mathcal{G}/\hbar}, \tag{44}$$
$$\hbar^{-2} F^{(3)} = \left( \frac{\tau_3}{9} - \frac{1}{2} \tau_1 \tau_2 \mathrm{D}\mathcal{G} + \tau_1^3 \left( \frac{1}{2} (\mathrm{D}\mathcal{G})^2 - \frac{\hbar}{6} \mathrm{D}^2 \mathcal{G} \right) \right) e^{-3\mathcal{G}/\hbar}.$$

In appendix B, we will also need the frame in which $\mathrm{D} = 0$, where the nonperturbative sectors reduce to

$$\hbar^{-2} F^{(n)} = \frac{(-1)^{n+1}}{n^2} \tau_n e^{-2\pi i n \mathcal{A}/\hbar}. \tag{45}$$

Note that in the latter frame, the transmonomials contain only the *classical period* $\mathcal{G} = \mathcal{A} = \beta t_0$, and so the nonperturbative sectors are not 'dressed' with asymptotic series of perturbative fluctuations in $\hbar$.

Now we arrive at the last and most important property of the quantum prepotential that we need: it was proposed in [24] that the pointed alien derivatives (see (13)) should act on the perturbative quantum prepotential as[11]

$$\dot{\Delta}_{l\mathcal{A}} F^{(0)} = S_{l\mathcal{A}} F_l^{(l)}, \tag{46}$$

where the $S_{l\mathcal{A}}$ are Stokes constants that are independent of $\hbar$, and

$$F_l^{(l)} = \frac{(-1)^{l+1}}{l^2} \hbar^2 e^{-l\mathcal{G}/\hbar}, \tag{47}$$

is simply $F^{(l)}$ with $\tau_l = 1$ and all other $\tau_k$ for $k \neq l$ set to zero. Subsequently, one can straightforwardly check that

$$\dot{\Delta}_{l\mathcal{A}} \mathcal{G} = S_{l\mathcal{A}} \hbar \frac{(-1)^l}{l} \mathrm{D}\mathcal{G} e^{-l\mathcal{G}/\hbar}, \tag{48}$$

using the fact that $\mathrm{D}$ and $\dot{\Delta}_{l\mathcal{A}}$ commute. Given equation (48) and the fact that all instanton corrections to the quantum prepotential are functions of $\mathcal{G}$, all pointed alien derivatives commute with eachother, i.e.

$$[\dot{\Delta}_{\mathcal{A}}, \dot{\Delta}_{\mathcal{A}'}] f(\mathcal{G}) = 0, \tag{49}$$

---

[11] This expression is adjusted to our normalisation (44).

for an arbitrary function $f(\mathcal{G})$.

Having completed our brief reviews of resurgence, the all-orders WKB method and the quantum prepotential, we are now ready to study the three quantum mechanical problems of the forthcoming sections. The techniques and concepts laid out in this subsection, together with a few more tools that we will introduce along the way, allow us to produce complete nonperturbative transseries solutions for the energy spectrum. By casting these structures in the language of alien calculus we can then obtain a deeper understanding of the Stokes phenomenon for energy spectra in quantum mechanics. As we will see shortly, these transseries are much more complicated than their cousins in the realm of ODEs. It is for this reason that we start with the example of the cubic oscillator, which from a resurgence point of view turns out to be the simplest of the three models, before advancing to the symmetric double well and cosine potentials.

# 3 The cubic oscillator

In order to demonstrate our approach we start with the cubic oscillator. This one-dimensional quantum mechanical model describes a particle in a cubic potential:

$$V(x) = \frac{x^2}{2} - x^3. \tag{50}$$

Although there is no stable vacuum in this model, the Schrödinger operator $\hat{H} = \hat{p}^2 + V(\hat{x})$ does admit a discrete spectrum of resonances when we impose Gamow-Siegert boundary conditions [38,39]: a decaying wave function as $x$ goes to $-\infty$ and a purely outgoing wave as $x$ goes to $+\infty$. When computing the spectrum of energies of these resonances – which are asymptotic divergent series in $\hbar$ – we learn that they are non-Borel summable, i.e. there are singularities on the positive real axis in the Borel plane, making the resummation procedure ambiguous. One way to circumvent this problem is to slightly deform $\hbar$ by giving it a small phase, moving the Borel singularities away from the real axis and rendering the series Borel-summable.

If we assume that the energy $E$ lies within the range $0 < E < \frac{1}{54}$, then all three turning points lie on the real axis, giving rise to a natural choice for the quantum A-period $t(E;\hbar)$ and the quantum B-period $t_D(E;\hbar)$, as displayed in figure 3. Using various techniques like e.g. instanton calculus [40], the Voros-Silverstone connection formula [5,41] or the uniform WKB method [, ], one can derive an *exact quantisation condition*: A functional relation, expressed in terms of the quantum periods, that allows us to compute the full nonperturbative energy spectrum of the resonances. More precisely, the quantisation condition fixes the value $t$ that the quantum A-period can attain in terms of an integer $N$, and therefore *quantises* the spectrum $E(t;\hbar)$, which we obtain after inverting the A-period. Regardless of the choice of computational method, the resulting quantisation condition will be ambiguous. When we slightly deform $\hbar$ as explained above, the ambiguity is resolved and as a result we have *two* quantisation conditions, one for $\text{Im}(\hbar) < 0$ and one for $\text{Im}(\hbar) > 0$. This therefore leads to two distinct asympotic expansions for the energy, which after resummation should be identical in the limit that the phase of $\hbar$ goes to zero. Understanding the Stokes transition that takes us from the former ($\text{Im}(\hbar) < 0$) to the latter expansion ($\text{Im}(\hbar) > 0$) will be the main objective of this section.

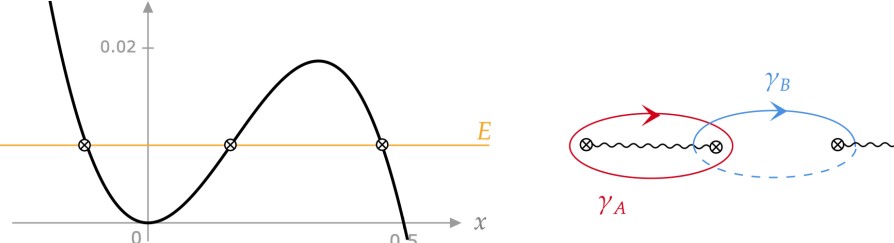

Figure 3: On the left the cubic potential (50) and on the right the same three turning points in the complex $x$-plane, with corresponding cycles $\gamma_A$ and $\gamma_B$ that one integrates over to obtain the A- and B-periods. The black wavy lines are square root branch cuts.

## 3.1 Iterative solution to the exact quantisation condition

In the case of the cubic oscillator, using the techniques mentioned above one derives the following two quantisation conditions for $\text{Im}(\hbar) < 0$ and $\text{Im}(\hbar) > 0$ respectively:

$$
\begin{aligned}
D^-(E;\hbar) &= 1 + e^{2\pi i t(E;\hbar)/\hbar} + e^{-t_D(E;\hbar)/\hbar} = 0\,, \\
D^+(E;\hbar) &= 1 + e^{2\pi i t(E;\hbar)/\hbar} = 0\,.
\end{aligned}
\tag{51}
$$

These conditions lead to two nonperturbatively different asymptotic solutions for the energy spectrum, whose difference should be accounted for by the Stokes phenomenon. The functions $D^+$ and $D^-$ used in the quantisation conditions defined above are functions of the independent variables $(E;\hbar)$. We now wish to change variables to $(t;\hbar)$. This is achieved by using the quantum mirror map $E(t;\hbar)$ to substitute $E$, leading to conditions that we can now write as

$$
\begin{aligned}
D^-(t;\hbar) &= 1 + e^{2\pi i t/\hbar} + e^{-t_D(t;\hbar)/\hbar} = 0\,, \\
D^+(t;\hbar) &= 1 + e^{2\pi i t/\hbar} = 0\,.
\end{aligned}
\tag{52}
$$

Note that now $t$ is an unknown which, after picking one of the two conditions, must be solved for. The quantisation condition $D^+(t;\hbar) = 0$ can readily be solved and yields

$$
t = \hbar\left(N + \frac{1}{2}\right),
\tag{53}
$$

for some nonnegative[12] integer $N$. This condition is called the *perturbative quantisation condition*, as it gives rise to the *perturbative* energy spectrum.

To better understand this statement, let us briefly digress from the main story and illustrate how (53) fixes the perturbative energy spectrum – see also [42]. The classical period $t_0(E)$ can be computed and yields

$$
t_0(E) = E \cdot {}_2F_1\left(\frac{1}{6},\frac{5}{6},2,54E\right) \simeq E + \frac{15E^2}{4} + \frac{1155E^3}{16} + \mathcal{O}(E^4).
\tag{54}
$$

Then using the differential operator technique[13] we can express the quantum corrections of the quantum A-period in terms of the classical period and its derivative with respect to $E$. The

---

[12]Note that we require $E > 0$ for the turning points to be real, otherwise a different Stokes graph with different quantisation conditions would emerge. Hence, consistency requires us to restrict $N$ to nonnegative integers $\mathbb{Z}_{\geq 0}$. In principle, at finite $\hbar$ there would also be an upper bound to $N$ where the potential is 'filled'; however, for us this upper bound plays no role since we will always work in a small $\hbar$-expansion.

[13]See e.g. [36,43] or appendix A of [44] for an explanation. In a nutshell, one uses the fact that the first cohomology group of a genus $g$ Riemann surface is spanned by $2g$ meromorphic differentials. The leading piece of the Riccati solution, $p_0(E,x)$, together with its derivatives with respect to $E$, form a suitable basis for this space and so we can decompose any higher order term $p_{n>0}(E,x)$ in this basis modulo total derivatives with respect to $x$.

first two corrections are given by

$$
t_1(E) = -\frac{5(1-108E)}{15E(1-54E)}\, t_0 + \frac{3}{4}\frac{\partial t_0}{\partial E}\,,
$$
$$
t_2(E) = \frac{7(4-633E+34182E^2)}{1536E^3(1-54E)^3}\, t_0 - \frac{7(1-108E)}{384E(1-54E)^2}\frac{\partial t_0}{\partial E}\,.
\tag{55}
$$

One then inverts the series by imposing that $E(t(E;\hbar);\hbar) = E$, which up to the same order in $\hbar$ implies that

$$
E_1(t) = -\frac{\partial E_0}{\partial t}t_1(E_0(t))\,,
$$
$$
E_2(t) = -\frac{1}{2}\frac{\partial^2 E_0}{\partial t^2}t_1(E_0(t))^2 - \frac{\partial E_0}{\partial t}t_2(E_0(t)) - \frac{\partial E_1}{\partial t}t_1(E_0(t))\,.
\tag{56}
$$

Computing these corrections and adding them up leads to a perturbative asymptotic expansion for $E$:

$$
\begin{aligned}
E(t;\hbar) = {} & \left(t - \frac{15t^2}{4} - \frac{705t^3}{16} - \frac{115755t^4}{128} + \mathcal{O}\left(t^5\right)\right)\\
& + \left(-\frac{7}{16} - \frac{1155t}{64} - \frac{209055t^2}{256} + \mathcal{O}\left(t^3\right)\right)\hbar^2\\
& + \left(-\frac{101479}{2048} + \mathcal{O}\left(t\right)\right)\hbar^4\\
& + \mathcal{O}(\hbar^6)\,.
\end{aligned}
\tag{57}
$$

Finally, if we let $\nu = N + \frac{1}{2}$, the perturbative energy after quantisation (53) becomes

$$
\begin{aligned}
E(\nu;\hbar) = {} & \nu\hbar - \left(\frac{7}{16} + \frac{15\nu^2}{4}\right)\hbar^2 - \left(\frac{1155\nu}{64} + \frac{705\nu^3}{16}\right)\hbar^3\\
& - \left(\frac{101479}{2048} + \frac{209055\nu^2}{256} + \frac{115755\nu^4}{128}\right)\hbar^4 + \mathcal{O}(\hbar^5)\,.
\end{aligned}
\tag{58}
$$

One can verify this expression, either by comparing with [ ] or by computing the energy levels using the Bender-Wu package [45] in Mathematica.

Returning to our main storyline, in the case of $D^+(t;\hbar) = 0$, the perturbative condition is the full quantisation condition, and we have now succesfully computed the resonance spectrum. However, when we cross the real positive $\hbar$-axis clockwise, the quantisation condition is altered: it obtains additional corrections which are nonperturbative in $\hbar$ – as shown in (52) when moving from the bottom to the top equation. The period $t$ which solves the condition $D^- = 0$ must therefore be different from the perturbative solution, and this difference can expressed in terms of a new nonperturbative quantity $\Delta t$ which promotes $t$ to a transseries $\hat{t}$ [ , ]:

$$
\hat{t} = t + \Delta t\,,\quad \text{with}\quad \Delta t = \sum_{n=1}^{\infty} t^{(n)}\lambda^n\,,
\tag{59}
$$

where $\lambda = \mathrm{e}^{-\frac{1}{\hbar}t_D(t;\hbar)}$ is an exponentially small quantity, which contains the instanton transmonomial and also a power series of its own. Note that here, we have temporarily gone back to the 'unquantised' situation where $t$ can be any value of the A-period for the '+'-resolution of the nonperturbative ambiguity, and we now want to express the A-period $\hat{t}$ for the '−'-resolution, including its nonperturbative contributions, in terms of $t$. From plugging the above Ansatz

into $D^- = 0$, we learn that in order to compute the instanton corrections we need to solve the equation

$$\Delta t = -\frac{i\hbar}{2\pi} \log\left(1 + e^{-\frac{1}{\hbar} t_D(t + \Delta t;\hbar)}\right). \tag{60}$$

We can solve this equation exactly, but before we do so, let us briefly recap how one solves the equation iteratively []. By plugging in the Ansatz from above one finds for the leading orders in $\lambda$ that

$$
\begin{aligned}
t^{(1)} &= -\frac{i\hbar}{2\pi}, \\
t^{(2)} &= \frac{i\hbar}{4\pi} + \frac{\hbar F''}{4\pi^2}, \\
t^{(3)} &= -\frac{i\hbar}{6\pi} - \frac{3\hbar F''}{8\pi^2} + \frac{3i\hbar(F'')^2}{16\pi^3} - \frac{i\hbar^2 F'''}{16\pi^3}.
\end{aligned} \tag{61}
$$

The $F$-derivatives in this expression are with respect to $t$; recall in particular from (34) that $F'(t;\hbar) = t_D(t;\hbar)$. Next, we can compute the energy transseries

$$\hat{E}(t;\hbar) = E(t + \Delta t;\hbar) = \sum_{n=0}^{\infty} E^{(n)} \lambda^n, \tag{62}$$

as an expansion in powers of $\lambda$ around its perturbative solution $E^{(0)} = E(t;\hbar)$, to obtain a full transseries. For the first few orders in $\lambda$ this yields

$$
\begin{aligned}
E^{(1)} &= -\frac{i\hbar E'}{2\pi}, \\
E^{(2)} &= \frac{i\hbar E'}{4\pi} + \frac{\hbar E' F''}{4\pi^2} - \frac{\hbar^2 E''}{8\pi^2}, \\
E^{(3)} &= -\frac{i\hbar E'}{6\pi} - \frac{3\hbar E' F''}{8\pi^2} + \frac{\hbar^2 E''}{8\pi^2} + \frac{3i\hbar E'\left(F''\right)^2}{16\pi^3} - \frac{i\hbar^2 E' F'''}{16\pi^3} - \frac{i\hbar^2 E'' F''}{8\pi^3} + \frac{i\hbar^3 E'''}{48\pi^3}.
\end{aligned} \tag{63}
$$

The nonperturbative corrections $t^{(n)}$ and $E^{(n)}$ above are expressed in terms of the 'perturbative' quantities $E$ and $F$, and are therefore also formal power series in $\hbar$ themselves. As we go to higher orders, even in the above form, the expressions for the instanton corrections (61) and (63) become larger and more unmanageable very fast. Fortunately, one can organise them in a more efficient way:

$$
\begin{aligned}
t^{(1)}\lambda &= -\frac{i\hbar}{2\pi}\lambda, \\
t^{(2)}\lambda^2 &= \frac{i\hbar}{4\pi}\lambda^2 - \frac{\hbar}{8\pi^2}\left(\hbar\frac{\partial}{\partial t}\lambda^2\right), \\
t^{(3)}\lambda^3 &= -\frac{i\hbar}{6\pi}\lambda^3 + \frac{\hbar}{8\pi^2}\left(\hbar\frac{\partial}{\partial t}\lambda^3\right) + \frac{i\hbar}{48\pi^3}\left(\hbar^2\frac{\partial^2}{\partial t^2}\lambda^3\right),
\end{aligned} \tag{64}
$$

and similarly for the energy transseries [, 42]:

$$
\begin{aligned}
E^{(1)}\lambda &= -\frac{i\hbar}{2\pi}E'\lambda, \\
E^{(2)}\lambda^2 &= \frac{i\hbar}{4\pi}E'\lambda^2 - \frac{\hbar}{8\pi^2}\hbar\frac{\partial}{\partial t}\left(E'\lambda^2\right), \\
E^{(3)}\lambda^3 &= -\frac{i\hbar}{6\pi}E'\lambda^3 + \frac{\hbar}{8\pi^2}\hbar\frac{\partial}{\partial t}\left(E'\lambda^3\right) + \frac{i\hbar}{48\pi^3}\hbar^2\frac{\partial^2}{\partial t^2}\left(E'\lambda^3\right).
\end{aligned} \tag{65}
$$

We have written these expressions using the 'dimensionless derivative' $\hbar \frac{\partial}{\partial t}$, which turns out to be very convenient. More generally, we would like to argue that the full transseries solutions $\hat{t}$ and $\hat{E}$ can be put in the following form:

$$\hat{t} = t + \hbar \sum_{n=1}^{\infty} \sum_{m=0}^{n-1} u_{n,m} \left( \hbar \frac{\partial}{\partial t} \right)^m \lambda^n, \tag{66}$$

and

$$\hat{E} = E + \hbar \sum_{n=1}^{\infty} \sum_{m=0}^{n-1} u_{n,m} \left( \hbar \frac{\partial}{\partial t} \right)^m \left( \frac{\partial E}{\partial t} \lambda^n \right). \tag{67}$$

Remarkably, the coefficients $u_{n,m}$ in both transseries are the exact same numbers – a pattern which is also clear in (64) and (65). In the next subsection we derive the form (66) and (67) explicitly, including exact expressions for these universal coefficients.

## 3.2 Deriving exact multi-instanton expressions

In [24], a solution to the holomorphic anomaly equations was presented which involved solving a functional equation using the Lagrange inversion theorem. Inspired by this result, we show that the same technique can be used to solve (60) and produce the coefficients $u_{n,m}$ of (66) and (67) in closed form. The resemblance between both approaches is not unexpected, given that the quantum prepotential – which solves the holomorphic anomaly equations in the NS background – is closely related to the ordinary energy through the PNP relation. We will introduce the PNP relation in the next subsection.

Consider a function $R(\Delta, z)$ of the generic form

$$R(\Delta; z) = \hbar \sum_{k=1}^{\infty} r_k z^k \exp \left( -\frac{k}{\hbar} e^{\Delta \partial_t} F'(t) \right), \tag{68}$$

where the $r_k$ are arbitrary coefficients and $e^{\Delta \partial_t}$ is a shift operator. One could take $F'(t)$ to also be an arbitrary function, but for our purposes it will always be the derivative of the quantum prepotential. The variable $z$ is a convenient bookkeeping device that we use to count powers of the instanton transmonomial. Let us for the moment forget about equation (60) and regard $\Delta t$ as the solution to the more generic equation

$$\Delta t = R(\Delta t; z). \tag{69}$$

This equation implies that the solution $\Delta t$ itself will also be an expansion in $z$. Now let us take an arbitrary function $\varphi(\Delta)$ that is analytic at $\Delta = 0$. Then the Lagrange inversion theorem (see e.g. theorem 2.4.1 of [46]) tells us that this function evaluated at $\Delta = \Delta t$, with $\Delta t$ the solution to (69), can be expressed in the following way:

$$\varphi(\Delta t) = \sum_{m=1}^{\infty} \frac{1}{m} [\Delta^{m-1}] \varphi'(\Delta) R(\Delta; z)^m. \tag{70}$$

Here $[\Delta^k]$ means that we extract the series coefficient associated to the $k$-th power of $\Delta$ from the expression that follows. Let us choose

$$\varphi(\Delta) = f(t + \Delta) = e^{\Delta \partial_t} f(t), \tag{71}$$

for some arbitrary function $f(t)$, infinitely differentiable at $t = 0$. Note again that the evaluation of $\varphi$ at $\Delta t$ induces a $z$ dependence which implies that we can write our solution as a power series

$$\varphi(\Delta t) = \sum_{n=0}^{\infty} f_n z^n, \tag{72}$$

for some yet unknown coefficients $f_n$. Below, we want to compute the $f_n$ and subsequently show that after removing the bookkeeping variable $z$ (by setting $z = 1$) we can write the full solution as

$$f(t + \Delta t) = f(t) + \hbar \sum_{n=1}^{\infty} \sum_{m=0}^{n-1} u_{n,m} \left( \hbar \frac{\partial}{\partial t} \right)^m \left( f'(t) \lambda^n \right), \tag{73}$$

and that we can produce *explicit* expressions for the coefficients $u_{n,m}$.

We start by noting that $f_0 = f(t)$, since (68) vanishes as $z$ goes to zero, and therefore so does $\Delta t$. The extraction $[\Delta^{m-1}]$ in equation (70) can in practice be done through differentiation, allowing us to write

$$f_n = \sum_{m=1}^{\infty} \frac{1}{m! n!} \left( \frac{\partial}{\partial z} \right)^n \left( \frac{\partial}{\partial \Delta} \right)^{m-1} \varphi'(\Delta) R(\Delta; z)^m \bigg|_{\Delta = z = 0}. \tag{74}$$

We first calculate[14]

$$\left( \frac{\partial}{\partial z} \right)^n R(\Delta; z)^m = \sum_{k=1}^{n} \left( \frac{\partial^k}{\partial R^k} R^m \right) B_{n,k}(R', R'', \dots), \tag{75}$$

using Faà di Bruno's formula, which generalises the chain rule to higher derivatives. The $B_{n,k}$ are incomplete Bell polynomials, with arguments that are derivatives of $R$ with respect to $z$. We then observe that these Bell polynomials are homogenous in the instanton transmonomial:[15]

$$B_{n,k}(R', R'', \dots) = \exp\left( -\frac{n}{\hbar} e^{\Delta \partial_t} F' \right) B_{n,k}(\tilde{R}', \tilde{R}'', \dots), \tag{76}$$

where we have extracted the instanton transmonomial from the $k$-th derivatives: $R^{(k)} = \exp\left( -\frac{k}{\hbar} e^{\Delta \partial_t} F' \right) \tilde{R}^{(k)}$. Subsequently, one applies the derivatives with respect to $\Delta$ and learns from a straightforward calculation that[16]

$$\left( \frac{\partial}{\partial \Delta} \right)^{m-1} \varphi'(\Delta) \exp\left( -\frac{n}{\hbar} e^{\Delta \partial_t} F' \right) \bigg|_{\Delta = 0} = \left( \frac{\partial}{\partial t} \right)^{m-1} f'(t) e^{-\frac{n}{\hbar} F'(t)}. \tag{77}$$

We then obtain

$$f_n = \frac{1}{n!} \sum_{k=1}^{n} \frac{\partial^k}{\partial R^k} \sum_{m=1}^{\infty} \frac{1}{m!} R^m \left( \frac{\partial}{\partial t} \right)^{m-1} f'(t) B_{n,k}\left( R'(0; z), R''(0; z), \dots \right) \bigg|_{z=0}$$

$$= \frac{1}{n!} \sum_{k=0}^{n-1} \left( \frac{\partial}{\partial t} \right)^k f'(t) B_{n,k+1}\left( R'(0; 0), R''(0; 0), \dots \right). \tag{78}$$

The above derivatives in the Bell polynomials are just instanton transmonomials: $R^{(k)}(0; 0) = k! \, r_k \hbar \, e^{-\frac{k}{\hbar} F'(t)}$, and so we end up with

$$f_n = \hbar \sum_{k=0}^{n-1} \frac{1}{n!} B_{n,k+1}\left( 1! \, r_1, 2! \, r_2, 3! \, r_3, \dots, (n-k)! \, r_{n-k} \right) \left( \hbar \frac{\partial}{\partial t} \right)^k f'(t) e^{-\frac{n}{\hbar} F'(t)}, \tag{79}$$

---

[14]Note that $\varphi(\Delta)$, which is evaluated at *generic* $\Delta$, is independent of $z$. It is only after evaluating it at $\Delta t$ that the function develops a $z$-dependence.

[15]We will call both $\exp\left( -\frac{1}{\hbar} F'(t) \right)$ and $\exp\left( -\frac{1}{\hbar} F'(t + \Delta t) \right)$ an 'instanton transmonomial'; it will generally be clear from the context which transmonomial we refer to.

[16]One uses the fact that

$$\left[ \frac{\partial^n}{\partial \Delta^n} e^{\Delta \partial_t} \right]_{\Delta=0} = \frac{\partial^n}{\partial t^n}.$$

Table 1: Values of the coefficients $u_{n,m}$ for the energy transseries (67). The index $n$ labels the rows and $m$ labels the columns.

| $m$ | 0 | 1 | 2 | 3 | 4 |
|---|---|---|---|---|---|
| $E^{(1)}$ | $\frac{1}{2\pi\mathrm{i}}$ | | | | |
| $E^{(2)}$ | $-\frac{1}{4\pi\mathrm{i}}$ | $-\frac{1}{8\pi^2}$ | | | |
| $E^{(3)}$ | $\frac{1}{6\pi\mathrm{i}}$ | $\frac{1}{8\pi^2}$ | $-\frac{1}{48\pi^3\mathrm{i}}$ | | |
| $E^{(4)}$ | $-\frac{1}{8\pi\mathrm{i}}$ | $-\frac{11}{96\pi^2}$ | $\frac{1}{32\pi^3\mathrm{i}}$ | $\frac{1}{384\pi^4}$ | |
| $E^{(5)}$ | $\frac{1}{10\pi\mathrm{i}}$ | $\frac{5}{48\pi^2}$ | $-\frac{7}{192\pi^3\mathrm{i}}$ | $-\frac{1}{192\pi^4}$ | $\frac{1}{3840\pi^5\mathrm{i}}$ |

which fits the form suggested in (73). Finally, we can then identify the coefficients

$$u_{n,m} = \frac{1}{n!} B_{n,m+1} \left( 1!\, r_1, 2!\, r_2, \dots, (n-m)!\, r_{n-m} \right). \tag{80}$$

Let us stress that other than infinite differentiability, this procedure makes no assumptions about $f(t)$ whatsoever: it simply tells us that if we shift its argument by the $\Delta t$ that solves (68), then we find the structure (73) with coefficients $u_{n,m}$ as given in (80). The values of these coefficients are thus independent of the details of the quantity $f(t)$; the approach is merely a recipe for enhancing the function $f(t)$ to the transseries $\hat{f}(t) = f(t + \Delta t)$.

Going back to our actual problem, we are perturbing the energy $E(t)$ with a $\Delta t$ that solves (60). Therefore we can identify $f(t) = E(t;\hbar)$ and compare (60) with (68) to find

$$r_k = \frac{(-1)^k}{k} \left( -\frac{1}{2\pi\mathrm{i}} \right). \tag{81}$$

Inserting this into (80) gives us the sought-for coefficients $u_{n,m}$ of our energy transseries; we tabulate some of these coefficients in table 1. Alternatively, when we choose $f(t) = t$ we obtain the transseries (66), for which we now understand why the same universal coefficients $u_{n,m}$ appear.

Having obtained explicit values for the universal coefficients $u_{n,m}$, we now have a complete description of the energy transseries solution for the cubic oscillator when $\hbar$ is located slightly below the real axis. The Stokes phenomenon that occurs along the ray $\arg(\hbar) = 0$ accounts for the appearance of all these instanton corrections, meaning that we should have the following action of the *Stokes automorphism*:

$$\mathfrak{S}_0 E(t;\hbar) = \hat{E}(t;\hbar). \tag{82}$$

On the left hand side we have the *perturbative* energy $E(t;\hbar)$ which is a solution to the quantisation condition $D^+ = 0$, and on the right hand side we find the *transseries* $\hat{E}(t;\hbar) = E(t+\Delta t;\hbar)$ which solves $D^- = 0$. We would like to fit this Stokes phenomenon into the more universal framework of *one-parameter transseries*. This involves introducing a new parameter $\sigma$ called the *transseries parameter* which incorporates the Stokes jump that our asymptotic solution for the energy undergoes. While being slightly agnostic for the moment about the precise details,

we thus enhance the notation of our energy transseries to include this parameter by writing $\hat{E}(t, \sigma; \hbar)$. The above Stokes jump should then translate to

$$\mathfrak{S}_0 \hat{E}(t, 0; \hbar) = \hat{E}(t, S; \hbar), \tag{83}$$

where $S$ is a yet unknown Stokes constant. It is then natural to expect that the coefficients $u_{n,m}$ encode this Stokes constant in some convoluted way. In order to extract the Stokes constant, we need to decompose the action of the Stokes automorphism in terms of pointed alien derivatives (cf. (12)), which are labeled by the locations of the singularities $l\mathcal{A} = lF_0'(t)$ in the Borel plane. Accordingly, acting with the automorphism on the perturbative energy yields for the first few orders:[17]

$$E + \underbrace{\dot{\Delta}_{\mathcal{A}} E}_{\text{1-instanton}} + \underbrace{\left(\frac{1}{2}\dot{\Delta}_{\mathcal{A}}^2 + \dot{\Delta}_{2\mathcal{A}}\right)E}_{\text{2-instanton}} + \underbrace{\left(\frac{1}{6}\dot{\Delta}_{\mathcal{A}}^3 + \dot{\Delta}_{2\mathcal{A}}\dot{\Delta}_{\mathcal{A}} + \dot{\Delta}_{3\mathcal{A}}\right)E}_{\text{3-instanton}} + \ldots = \hat{E}(t, S; \hbar), \tag{84}$$

which should match the transseries below the Stokes line. Recall that the pointed alien derivative $\dot{\Delta}_{l\mathcal{A}}$ by definition (13) contributes a factor $e^{-l\mathcal{A}/\hbar}$ and hence we can group the terms together as shown above. If we can now figure out how the individual pointed alien derivatives act on the perturbative energy $E(t; \hbar)$, then we should be able to extract the Stokes constant $S$. In order to achieve this, we turn to the *PNP relation*.

## 3.3 Stokes phenomenon through the PNP relation

Guided by observations made in [47] while studying the Stark effect Hamiltonian, it was recognised in [] that for the cubic oscillator there exists a simple equation relating the tunneling period $t_D(t; \hbar)$ to the perturbative energy $E(t; \hbar)$. This equation was subsequently also established in various other quantum mechanical models including the double well [, 9] and cosine potentials [27] that we will study shortly. In fact, one can derive it for generic polynomial potentials by applying an appropriate Riemann bilinear identity on the WKB curve, as we illustrate in appendix A. This relation, which is often referred to as the PNP relation,[18] has appeared in the literature in many forms and guises. In our present setting and notation, it allows us to connect the ordinary energy $E$ to the quantum prepotential $F$ from (33) as

$$c\frac{\partial E}{\partial t} = t\frac{\partial^2 F}{\partial t^2} - \frac{\partial F}{\partial t} + \hbar\frac{\partial^2 F}{\partial t \partial \hbar}. \tag{85}$$

Here, $c$ is a constant of proportionality that one can compute explicitly, though its actual value will not be important for our purposes. An alternative expression is obtained by integrating the equation with respect to $t$, leading to

$$cE = \left(t\frac{\partial}{\partial t} - 2 + \hbar\frac{\partial}{\partial \hbar}\right)F + C(\hbar). \tag{86}$$

The integration constant $C(\hbar)$ can also be computed for each given model, and turns out to be a monomial[19] in $\hbar$. We provide more background information on this equation in appendix A. Let us stress that the PNP relation (86) is fundamentally different from the exact quantisation condition (51), despite the fact that both involve $E$ and $F$. The quantisation condition tells

---

[17]Formally, the 3-instanton correction contains the term $\frac{1}{2}(\dot{\Delta}_{2\mathcal{A}}\dot{\Delta}_{\mathcal{A}} + \dot{\Delta}_{\mathcal{A}}\dot{\Delta}_{2\mathcal{A}})$, but as we argued in section 2.1 the pointed alien derivatives commute.

[18]The name PNP stands for 'perturbative/nonperturbative' and comes from the fact that the relation connects the tunneling period – which induces nonperturbative corrections – to the perturbative energy $\hbar$-expansion.

[19]In fact, dimensional analysis of the Schrödinger equation tells us that if $\hbar$ has mass dimension 1, then the energy and quantum prepotential have mass dimension 2, which implies that $C(\hbar) \sim \hbar^2$.

us how the energy *transseries* $\hat{E}$ can be constructed from the *perturbative* energy $E$ and the *perturbative* quantum prepotential $F$, without making any assumptions about the specifics of these series expansions themselves. The PNP relation actually relates the two perturbative series to one another.

Our strategy is now as follows: for the quantum prepotential we have a clear understanding of its transseries solution and resurgent structure. Using the action of the pointed alien derivative on this quantum prepotential – given in (46) as proposed in [24] – we can use the PNP relation to probe the resurgent structure of the energy transseries. Formally what we would like to do, in the spirit of [21] (and also [23]), is therefore to extend the PNP relation from an equation between formal power series to an equation between transseries:

$$c\hat{E} = \left( t\frac{\partial}{\partial t} - 2 + \hbar\frac{\partial}{\partial \hbar} \right)\hat{F} + C(\hbar). \tag{87}$$

$\hat{E}$ and $\hat{F}$ now denote the respective transseries extensions of the energy and quantum prepotential. Because the energy transseries admits an expansion in integer powers of $\lambda$, the quantum prepotential transseries, whose first three corrections are given in (44), must have $\mathcal{G} = F'(t;\hbar)$ in order for the above equation to make sense. Starting from the PNP relation and applying the pointed alien derivative with $\mathcal{A} = t_{D,0}(t)$ to the *perturbative* energy $E$ we get

$$\begin{aligned}
\dot{\Delta}_{\mathcal{A}}E &= \frac{1}{c}\left( t\frac{\partial}{\partial t} - 2 + \hbar\frac{\partial}{\partial \hbar} \right)\dot{\Delta}_{\mathcal{A}}F \\
&= \frac{S_{\mathcal{A}}\hbar}{c}\left( -tF'' + F' - \hbar\frac{\partial F'}{\partial \hbar} \right)e^{-F'/\hbar} \\
&= -S_{\mathcal{A}}\hbar E'\lambda.
\end{aligned} \tag{88}$$

Here, we used the fact that the pointed alien derviative $\dot{\Delta}_{\mathcal{A}}$ commutes with differentiation with respect to $\hbar$ or $t$. In the final expression, we recognise the one-instanton correction from (65), which allows us to read off the value of the Stokes constant:

$$S_{\mathcal{A}} = -\frac{1}{2\pi i}. \tag{89}$$

We can repeat this computation for the pointed alien derivative with arbitrary 'step size' $l\mathcal{A}$ for $l \in \mathbb{Z}_{>0}$. Moreover, in anticipation of what is to come, it will be useful to apply the pointed alien derivatives to the instanton transmonomial $\lambda$ as well. We then find the following two similar-looking equations:

$$\begin{aligned}
\dot{\Delta}_{l\mathcal{A}}E &= \frac{S_{l\mathcal{A}}\hbar(-1)^l}{l}\frac{\partial E}{\partial t}\lambda^l, \\
\dot{\Delta}_{l\mathcal{A}}\lambda &= \frac{S_{l\mathcal{A}}\hbar(-1)^l}{l}\frac{\partial \lambda}{\partial t}\lambda^l.
\end{aligned} \tag{90}$$

Now, let us define

$$E_{n,m} = \left( \hbar\frac{\partial}{\partial t} \right)^m\left( \frac{\partial E}{\partial t}\lambda^n \right), \tag{91}$$

which are the essential building blocks of our exact transseries (67) that there are weighed by the coefficients $u_{n,m}$. Then, by combining equations (90) and (91) with the Leibniz rule for operators, we obtain

$$\dot{\Delta}_{l\mathcal{A}}E_{n,m} = \frac{S_{l\mathcal{A}}(-1)^l}{l}E_{n+l,m+1}, \tag{92}$$

which can alternatively be derived using the PNP relation similar to what we did in (88). This relation describes the full resurgent structure of the energy transseries in a nutshell, as

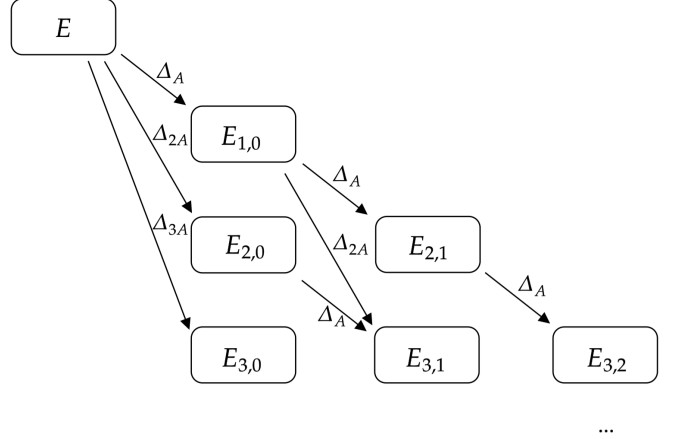

Figure 4: A pictorial representation of the result (92) which is also known as the *alien lattice* of the ordinary energy transseries $\hat{E}$. The boxes represent the building blocks (91) with the inclusion of the perturbative energy $E$, and the arrows illustrate how the pointed alien derivatives map each sector to one appearing in the next column.

depicted graphically in figure 4, and is a direct consequence of (46). From comparing our formula (92) to the coefficients $u_{n,m}$ in table 1, we learn that all Stokes constants reduce to one and the same number:

$$S_{l\mathcal{A}} = -\frac{1}{2\pi i} \, . \tag{93}$$

Remarkably, we now see that the condensed form (67) of writing the transseries is not only favorable from a notational point of view, but it also exposes its resurgent structure: if we assign a weight $l$ to the pointed alien derivative $\Delta_{l\mathcal{A}}$, then *every* sector $E_{n,m}$ in the transseries that occurs above the Stokes line is generated by a sum of strings of $m$ pointed alien derivatives with a total weight $n$ that naturally appears when we expand the automorphism $\mathfrak{S}_0$.

Summarizing what we have learned so far, we can now reformulate the energy transseries in the form of a more generic one-parameter transseries:

$$\hat{E}(t,\sigma;\hbar) = E(t;\hbar) + \hbar \sum_{n=1}^{\infty} \sum_{m=0}^{n-1} u_{n,m} \, \sigma^{m+1} \left( \hbar \frac{\partial}{\partial t} \right)^m \left( \frac{\partial E}{\partial t}(t;\hbar) \lambda(t;\hbar)^n \right), \tag{94}$$

with the universal coefficients $u_{n,m}$ given by (80), where

$$r_k = \frac{(-1)^k}{k} \, . \tag{95}$$

Notice how this transseries structure is different from the conventional ODE type one-parameter transseries – see e.g. [30, 48] – where the $n$-th instanton sector only carries a factor $\sigma^n$. Here, instead, we see that each instanton sector $E^{(n)}$ consists of multiple parts, each scaling with a different power of the transseries parameter, running from $\sigma^1$ up to $\sigma^n$. The complete Stokes transition can now be captured in the form (83) by the single expression

$$\mathfrak{S}_0 \hat{E}(t,0;\hbar) = \hat{E}\left(t, -\frac{1}{2\pi i}; \hbar\right) . \tag{96}$$

In fact, from the alien calculus structure uncovered above, we can argue that more generally when we cross the Stokes line we have

$$\mathfrak{S}_0 \hat{E}(t,\sigma;\hbar) = \hat{E}(t, \sigma + S_{\mathcal{A}}; \hbar), \tag{97}$$

for the same transseries with generic $\sigma$. Let us emphasise that the Stokes constants $S_{l\mathcal{A}}$ are first and foremost the Stokes constants of the quantum prepotential, defined in (46). We have merely picked a convenient normalisation for the generic energy transseries (94), with the redefinition of the $r_k$ given in (95), such that its Stokes constants are exactly the same as those of the quantum prepotential. This is possible thanks to the linear nature of the PNP relation.

Finally, let us remark that in this example *all* nonperturbative sectors $E_{n,m}$ are connected to the perturbative energy $E$ through resurgence, as shown in figure 4. This means that from studying the large order behaviour of the perturbative coefficients one can reconstruct the full transseries solution with the exception of the value of the transseries parameter $\sigma$. Therefore, the energy transseries $\hat{E}$ that we have constructed is a minimal resurgent transseries. As we will see shortly, this does not hold for the energy transseries that we find in potentials with multiple minima.

### 3.4 Stokes phenomenon from DDP formula

The alien calculus (92) uncovered above, including the value of the Stokes constants, can also be derived from the Delabaere-Dillinger-Pham formula directly. Let us illustrate this. We first switch back to the formalism in which the periods are expressed in terms of the independent variables $(E, \hbar)$. The DDP formula (31) along the $\arg(\hbar) = 0$ Stokes ray, expressed for both periods, then reads:

$$
\begin{aligned}
\mathfrak{S}_0 t(E; \hbar) &= t(E; \hbar) + \frac{i\hbar}{2\pi} \log\left(1 + e^{-t_D(E; \hbar)/\hbar}\right), \\
\mathfrak{S}_0 t_D(E; \hbar) &= t_D(E; \hbar).
\end{aligned}
\tag{98}
$$

The first of these two equations allows us to deduce the action of the pointed alien derivative on the quantum A-period:

$$
\dot{\Delta}_{l\mathcal{A}} t(E; \hbar) = \left(-\frac{1}{2\pi i}\right) \frac{\hbar(-1)^{l+1}}{l} e^{-l\, t_D(E; \hbar)/\hbar}.
\tag{99}
$$

To derive the action of the pointed alien derivative on the energy, we use the fact that the energy $E(t; \hbar)$ is the inverse of the quantum A-period $t(E; \hbar)$:

$$
0 = \dot{\Delta}_{l\mathcal{A}} E(t(E; \hbar); \hbar) = \dot{\Delta}_{l\mathcal{A}} E(t; \hbar)\Big|_{t=t(E;\hbar)} + \frac{\partial E}{\partial t}(t; \hbar)\Big|_{t=t(E;\hbar)} \dot{\Delta}_{l\mathcal{A}} t(E; \hbar),
\tag{100}
$$

where in the second equality we used the chain rule (18). Combining the previous two equations, we get

$$
\dot{\Delta}_{l\mathcal{A}} E(t; \hbar)\Big|_{t=t(E;\hbar)} = \frac{\partial E}{\partial t}(t; \hbar)\Big|_{t=t(E;\hbar)} \left(-\frac{1}{2\pi i}\right) \frac{\hbar(-1)^{l}}{l} e^{-l\, t_D(E; \hbar)/\hbar}.
\tag{101}
$$

We then switch to independent variables $(t, \hbar)$ by substituting $E$ by its quantum mirror map $E(t; \hbar)$ and obtain

$$
\dot{\Delta}_{l\mathcal{A}} E = \left(-\frac{1}{2\pi i}\right) \frac{\hbar(-1)^{l}}{l} \frac{\partial E}{\partial t} \lambda^{l}.
\tag{102}
$$

This is exactly the first equation of (90) with the Stokes constant $S_{l\mathcal{A}} = -\frac{1}{2\pi i}$. To obtain the second equation, we consider the second line of (98), which tells us that the dual quantum period $t_D(E; \hbar)$ is invariant under the Stokes jump. Given the identity (36) that expresses $t_D$ in terms of the quantum prepotential, and using the chain rule again, we obtain

$$
(\dot{\Delta}_{l\mathcal{A}} F')(t; \hbar)\Big|_{t=t(E;\hbar)} = F''(t; \hbar)\Big|_{t=t(E;\hbar)} \left(-\frac{1}{2\pi i}\right) \frac{\hbar(-1)^{l}}{l} e^{-l\, t_D(E; \hbar)/\hbar},
\tag{103}
$$

where the primes denote differentiation with respect to $t$. We then substitute $E$ by its quantum mirror map $E(t;\hbar)$, to obtain

$$\dot{\Delta}_{l\mathcal{A}}F'(t;\hbar) = F''(t;\hbar)\left(-\frac{1}{2\pi\mathrm{i}}\right)\frac{\hbar(-1)^l}{l}\mathrm{e}^{-l\,t_D(t;\hbar)/\hbar}. \tag{104}$$

Finally, using this result we straightforwardly derive the statement

$$\dot{\Delta}_{l\mathcal{A}}\lambda = \left(-\frac{1}{2\pi\mathrm{i}}\right)\frac{\hbar(-1)^l}{l}\frac{\partial\lambda}{\partial t}\lambda^l, \tag{105}$$

which is the second equation of (90) with the same Stokes constant $S_{l\mathcal{A}} = -\frac{1}{2\pi\mathrm{i}}$. Both equations in (90) imply the result (92) for $E_{n,m}$, as we argued earlier, and so we once again end up with the alien lattice displayed in figure 4.

Let us remark that the same approach can be used to derive relations (46) for the quantum prepotential along *any* direction in the complex $\hbar$-plane. We give two examples of this in appendix B.

## 3.5  On the quantum prepotential transseries

We conclude this section by calculating the quantum prepotential transseries that corresponds to the energy transseries (67), and generalise it to a one-parameter quantum prepotential transseries just like we did for the energy. There are two ways of doing this. The first one is by mapping instanton sectors of the quantum prepotential transseries (44) to the energy transseries via the nonperturbative PNP relation (87) – see e.g. [23]. One can then fix the parameters $\tau_k$ of the quantum prepotential transseries (44) by comparison to the coefficients $u_{n,m}$ of the energy transseries. As we did for the energy, one can then generalise this quantum prepotential transseries to a one-parameter transseries. The second option is to use the algorithm of section 3.2 to directly obtain a transseries expression for the quantum prepotential, and compare it to the solution (44) found in [24]. Here, we illustrate the latter approach.

We start by recalling that our algorithm requires us to pick a function $f(t)$ which we wish to perturb by shifting $t$ to $t + \Delta t$. Since the quantum prepotential is defined in terms of the dual quantum period by the relation (34), $F'(t;\hbar) = t_D(t;\hbar)$, a suitable candidate for $f(t)$ is to use the dual quantum period. We set $\varphi(\Delta) = f(t + \Delta) = F'(t + \Delta;\hbar)$ and obtain the transseries

$$\hat{F}'(t;\hbar) = F'(t;\hbar) + \hbar\sum_{n=1}^{\infty}\sum_{m=0}^{n-1}u_{n,m}\left(\hbar\frac{\partial}{\partial t}\right)^m\left(F''(t;\hbar)\lambda(t)^n\right), \tag{106}$$

in terms of the universal coefficients $u_{n,m}$ (80). Notice that $\lambda(t) = \exp(-F'(t;\hbar)/\hbar)$ and therefore we can integrate both sides with respect to $t$ to obtain

$$\hat{F} = F + \hbar^2\sum_{n=1}^{\infty}\left(-\frac{1}{n}u_{n,0}\lambda^n + \sum_{m=1}^{n-1}u_{n,m}\left(\hbar\frac{\partial}{\partial t}\right)^{m-1}\left(F''\lambda^n\right)\right). \tag{107}$$

Note that any integration constant must be $t$-independent and hence must be part of the perturbative quantum prepotential $F$. Comparing the above transseries to the solution given in [24], normalized as in (44), allows us to relate the coefficients $u_{n,m}$ to the parameters $\tau_k$ of the quantum prepotential. We show the result in table 2. Plugging in the actual values of the coefficients $u_{n,m}$ – that is, using the original $r_k$ from (81) – then tells us that all parameters $\tau_k$ have the same value:

$$\tau_k = k\,(-1)^k\,r_k = -\frac{1}{2\pi\mathrm{i}}. \tag{108}$$

Table 2: The coefficients $u_{n,m}$, expressed in terms of the parameters $\tau_k$ (44) of the quantum prepotential transseries $\hat{F}$ for the cubic oscillator.

| $m$ | 0 | 1 | 2 | 3 | 4 |
|---|---|---|---|---|---|
| $F^{(1)}$ | $-\tau_1$ | | | | |
| $F^{(2)}$ | $\frac{\tau_2}{2}$ | $\frac{\tau_1^2}{2}$ | | | |
| $F^{(3)}$ | $-\frac{\tau_3}{3}$ | $-\frac{\tau_2\tau_1}{2}$ | $-\frac{\tau_1^3}{6}$ | | |
| $F^{(4)}$ | $\frac{\tau_4}{4}$ | $\frac{\tau_3\tau_1}{3}+\frac{\tau_2^2}{8}$ | $\frac{\tau_2\tau_1^2}{4}$ | $\frac{\tau_1^4}{24}$ | |
| $F^{(5)}$ | $-\frac{\tau_5}{5}$ | $-\frac{\tau_4\tau_1}{4}-\frac{\tau_3\tau_2}{6}$ | $-\frac{\tau_3\tau_1^2}{6}-\frac{\tau_2^2\tau_1}{8}$ | $-\frac{\tau_2\tau_1^3}{12}$ | $-\frac{\tau_1^5}{120}$ |

If we introduce a transseries parameter by replacing the coefficients $u_{n,m}$ in (106) by $u_{n,m}\sigma^{m+1}$ and use the new $r_k$ from (95), similarly to what we did for the energy transseries, then we can identify $\sigma$ as the single transseries parameter of the quantum prepotential transseries, i.e.

$$\tau_k = \sigma. \tag{109}$$

We can then write the quantum prepotential transseries as

$$\hat{F}(t,\sigma;\hbar) = F + \hbar^2 \sum_{n=1}^{\infty}\left[ -\frac{1}{n}u_{n,0}\sigma\lambda^n + \sum_{m=1}^{n-1}u_{n,m}\sigma^{m+1}\left(\hbar\frac{\partial}{\partial t}\right)^{m-1}\left(F''\lambda^n\right)\right], \tag{110}$$

where the universal coefficients $u_{n,m}$ from (80) are defined using the new $r_k$ coefficients (95). The Stokes phenomenon of the quantum prepotential is then captured by

$$\mathfrak{S}_0\hat{F}(t,\sigma;\hbar) = \hat{F}(t,\sigma+S_{\mathcal{A}};\hbar). \tag{111}$$

Looking back, we can see that the resurgent structure of both the energy and the quantum prepotential are essentially the same due to the PNP relation. The singularities in their respective Borel planes naturally lead to the minimal transseries extensions (94) and (110) which carry a *single* degree of freedom $\sigma$ governing the nonperturbative effects. The exact quantisation conditions $D^+ = 0$ and $D^- = 0$ are then physical boundary conditions which fix the value of this parameter to $\sigma = 0$ and $\sigma = -\frac{1}{2\pi i}$ respectively. Furthermore, for both energy and quantum prepotential we can deduce the alien lattice structure shown in figure 4 and the exact values for the Stokes constants (93). These follow from translating the known alien calculus (46) of the quantum prepotential to the energy transseries and attributing the ambiguities that come from the quantisation conditions to the Stokes constants. Alternatively, one can reach the same conclusions directly from the Delabaere-Dillinger-Pham formula.

With that, we conclude the treatment of the cubic potential and move on to the double well potential, whose energy transseries, as we will see shortly, has an extra layer of complexity to it: its energy transseries will be no longer minimal, but will be a *sum* of minimal transseries.

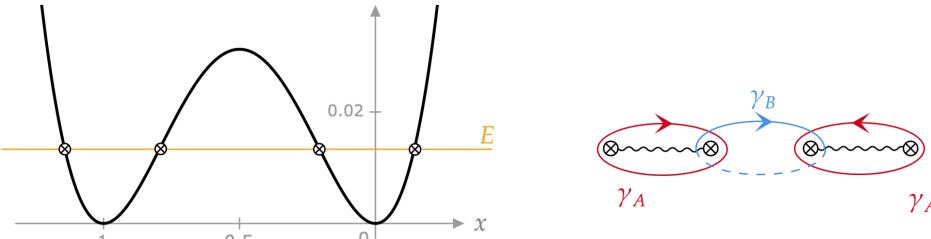

Figure 5: On the left we draw the symmetric double well potential (112), and on the right its four turning points in the complex $x$-plane with the cycles $\gamma_A$ and $\gamma_B$.

## 4   The symmetric double well potential

In this section, we consider the symmetric double well potential:

$$V(x) = \frac{x^2}{2}(1+x)^2\,. \tag{112}$$

When the energy $E$ lies in the range $0 < E < \frac{1}{32}$, there are four turning points on the real axis. These provide us with three cycles – see also figure 5: two cycles around minima that we both label by $\gamma_A$, since by symmetry they give rise to the same quantum A-period $t$, and the 'tunneling cycle' $\gamma_B$ that we associate to the dual quantum period $t_D$. Contrary to the cubic oscillator, this model does admit bound states. The goal will be to cast its nonperturbative spectrum of energies $E$ in the form of an exact transseries and study its Stokes phenomenon.

The exact quantisation condition for this model was obtained in [3] through instanton calculus, and can also be derived using the Voros-Silverstone connection formula [4, 41] (see also [42]) or the uniform WKB method [, 10]. As was the case for the cubic oscillator, each of these approaches leads to an ambiguity in the quantisation condition, which can be resolved by slightly perturbing the phase of $\hbar$. As a result, we are presented with two exact quantisation conditions that we write as $D^- = 0$ for $\text{Im}(\hbar) < 0$ and $D^+ = 0$ for $\text{Im}(\hbar) > 0$. In terms of independent variables $(t, \hbar)$ these conditions read

$$D_\epsilon^\pm = 1 + e^{\pm 2\pi i t/\hbar} \mp i\epsilon\, e^{-\frac{1}{2\hbar} t_D(t;\hbar)} = 0\,. \tag{113}$$

There are two novelties compared to the exact quantization conditions (52) of the cubic. First of all there are now *two* pairs of quantization conditions, labeled by a parameter $\epsilon \in \{+1, -1\}$, which refers to the parity of the wave function for which we compute the energy, and which can be interpreted as a quantum number. Almost all quantities that we compute will depend on a choice of $\epsilon$, but to avoid cluttering the notation with subscripts we generally suppress this dependence. Secondly, we note the subtle factor $\frac{1}{2}$ in (113) that multiplies the dual quantum period $t_D$. This factor indicates the presence of nonperturbative corrections that we associate with instantons that tunnel an *odd*[20] number of times.

Now, we follow a similar approach as for the cubic oscillator (52). We start by noting that the contribution of the dual 'tunneling' quantum period is exponentially suppressed along the positive real $\hbar$-axis, and we therefore first consider the *perturbative* quantisation condition

$$1 + e^{\pm 2\pi i t/\hbar} = 0\,, \quad \text{implying} \quad t = \hbar\left(N + \frac{1}{2}\right)\,. \tag{114}$$

---

[20]Note that the cubic only has corrections coming from instantons that tunnel an *even* number of times. This can be explained by the fact that the cubic has only a single local minimum which forces any instanton solution to tunnel back.

where $N$ is an integer. Note that the perturbative quantisation condition is insensitive to the choice of parity $\epsilon$, and therefore the perturbative energy spectrum, labeled by the quantum numbers $(\epsilon, N)$, is doubly degenerate. The next step is then to look for the fully *nonperturbative* solution $\hat{t}$ to (113), which we write as

$$\hat{t} = t + \Delta t, \quad \text{with} \quad \Delta t = \sum_{n=1}^{\infty} t^{(n)} \lambda^{\frac{1}{2}n}, \tag{115}$$

where $\lambda = \exp(-t_D(t;\hbar)/\hbar)$ as before, but due to the factor of $\frac{1}{2}$ in front of $t_D$ in (113) we now make an Ansatz in *half-integer* powers of this $\lambda$. This difference with the cubic case will play a central role in much of what follows. One can now show that this Ansatz produces a transseries for $\hat{t}$ which is found by solving the equation

$$\Delta t = \mp \frac{i\hbar}{2\pi} \log\left(1 \mp i\epsilon\, e^{-\frac{1}{2\hbar} t_D(t+\Delta t)}\right), \tag{116}$$

where the signs depend on whether we choose to solve $D^+ = 0$ or $D^- = 0$. Here and below, we will follow the convention that the upper sign in expressions is the one for the quantisation condition $D^+ = 0$, and the lower sign is the one for $D^- = 0$. Note that the equation above also has a factor $\frac{1}{2}$ in front of the dual quantum period $t_D$ in the exponent. Therefore our Ansatz for the energy transseries will also be one in powers of $\lambda^{\frac{1}{2}}$:

$$\hat{E} = E + \hbar \sum_{n=1}^{\infty} \sum_{m=0}^{n-1} u_{n,m} \left(\hbar \frac{\partial}{\partial t}\right)^m \left(\frac{\partial E}{\partial t} \lambda^{\frac{1}{2}n}\right). \tag{117}$$

We then turn to the algorithm that we introduced in the previous section and compute the universal coefficients $u_{n,m}$ given by (80), with input data

$$r_k = \frac{\epsilon^k (\pm i)^{k+1}}{k} \frac{1}{2\pi}. \tag{118}$$

These values follow from expanding (116) and comparing it to the expression

$$\Delta = R(\Delta; z) = \hbar \sum_{k=1}^{\infty} r_k z^k \exp\left(-\frac{k}{2\hbar} e^{\Delta \partial_t} F'(t)\right), \tag{119}$$

which is almost identical to (68), except again for the factor $\frac{1}{2}$ in the exponent. This yields the universal coefficient values displayed in table 3, the first three rows of which were also computed in [10] (see also [42]).

A few remarks are in order. First of all, we note that all the odd instanton sectors $E^{(2k+1)}$ depend on the parity $\epsilon$ whereas the even sectors do not. The odd sectors therefore induce a nonperturbative level-splitting that lifts the degeneracy of the perturbative energy levels. From a resurgence point of view, this also has an interesting implication. The insensitivity to the parity of the *even* instanton sectors (including the perturbative sector $E^{(0)}$) implies that from a resurgence point of view, one cannot deduce the existence of the *odd* sectors from the large order behaviour of the perturbative coefficients. This is a well known fact which is graphically depicted in the the resurgence triangle of [26, 27]. We will elaborate more on this when we study the cosine potential in the next section.

Secondly, it is well known [13] that the large order behaviour of the perturbative energy to leading order probes the imaginary 2-instanton effect that comes from an *instanton-anti-instanton pair*. Such an imaginary term does appear in table 3 at the 2-instanton level and is also ambiguous – i.e. dependent on the $\pm$ quantisation condition chosen to resolve the non-Borel summability of the perturbative series. In fact, all imaginary parts of the transseries

Table 3: The values for the coefficients $u_{n,m}$ of the energy transseries (117). The index $n$ labels the rows and $m$ labels the columns. In tabulating these values we used $(\pm)^2 = \epsilon^2 = 1$.

| $m$ | 0 | 1 | 2 | 3 | 4 |
|---|---|---|---|---|---|
| $u_{1,m}$ | $-\frac{1}{2\pi}\epsilon$ | | | | |
| $u_{2,m}$ | $\mp\frac{i}{4\pi}$ | $\frac{1}{8\pi^2}$ | | | |
| $u_{3,m}$ | $\frac{1}{6\pi}\epsilon$ | $\pm\frac{i}{8\pi^2}\epsilon$ | $-\frac{1}{48\pi^3}\epsilon$ | | |
| $u_{4,m}$ | $\pm\frac{i}{8\pi}$ | $-\frac{11}{96\pi^2}$ | $\mp\frac{i}{32\pi^3}$ | $\frac{1}{384\pi^4}$ | |
| $u_{5,m}$ | $-\frac{1}{10\pi}\epsilon$ | $\mp\frac{5i}{48\pi^2}\epsilon$ | $\frac{7}{192\pi^3}\epsilon$ | $\pm\frac{i}{192\pi^4}\epsilon$ | $-\frac{1}{3840\pi^5}\epsilon$ |

are ambiguous in this way, as is evident from equation (116), and one would expect that when resumming the whole transseries these ambiguities cancel the ambiguities that arise from resummming individual power series in the transseries. This expectation is supported by the fact that the double well is a confining potential and therefore its energy spectrum should be real and unambiguous. This cancellation mechanism has been tested in various works [14, 15] (see also [42]) and we would like to extend it to all orders by casting it in the language of alien calculus.

## 4.1 The full transseries

We would now like to probe the resurgent structure of the energy transseries (117) using alien calculus and formulate a generic transseries structure as we did in (94) for the cubic. In the double well potential, this is a more delicate exercise because its energy transseries is an expansion in *half-integer* powers of $\lambda$ whereas the singularities in the Borel plane of the perturbative energy are found at integer multiples of the primitive action $\mathcal{A} = t_{D,0}(E)$, which therefore correspond to *integer* powers of $\lambda$. Hence, there are *two* different transseries that we can identify: One that includes *all* nonperturbative sectors, and which we call the *full* transseries, and one that only contains those sectors visible in the Borel plane of the perturbative series, called the *minimal* transseries.

The energy transseries (117) with the coefficients shown in table 3 is an example of a full transseries, and we will construct the corresponding full quantum prepotential transseries in this subsection. The *minimal* transseries is, as the name suggests, a truly minimal resurgent transseries as defined in section 2.1. It allows us to derive the Stokes data that were obtained in [24] using large order methods, and it admits the same generalisation to a one-parameter transseries as the energy and quantum prepotential found for the cubic potential. Understanding how one obtains these minimal transseries and how they relate to the full transseries will be the topic of the subsequent subsections in our treatment of the double well potential.

Let us therefore start by deriving the *full* quantum prepotential transseries. First, we recall the nonperturbative extension (87) of the PNP relation. Our full energy transseries (117) relates through this PNP relation to the full quantum prepotential transseries $\hat{F}^{\text{full}}$, which there-

fore must have the following form:

$$\hat{F}^{\text{full}} = \sum_{n=0}^{\infty} \tilde{F}^{(n)}, \quad \text{with} \quad \tilde{F}^{(n)} \sim \lambda^{\frac{1}{2}n}. \tag{120}$$

We can construct this transseries using the same approach as in section 3.5. Given $\Delta t$ which solves equation (116), we again choose $\varphi(\Delta t) = f(t + \Delta t) = F'(t + \Delta t; \hbar)$ and use our algorithm to produce a transseries for the dual quantum period in integer powers of $\lambda^{\frac{1}{2}}$. We then integrate both sides with respect to $t$ and obtain an expression for the full quantum prepotential transseries:

$$\hat{F}^{\text{full}} = F + \hbar^2 \sum_{n=1}^{\infty} \left[ -\frac{2}{n} u_{n,0} \lambda^{\frac{1}{2}n} + \sum_{m=1}^{n-1} u_{n,m} \left( \hbar \frac{\partial}{\partial t} \right)^{m-1} F'' \lambda^{\frac{1}{2}n} \right], \tag{121}$$

which is only slightly different from its cubic analogue (107) by a few factors of two. We can then identify this transseries with the transseries structure from [24] that we wrote in (44), where we have $\mathsf{D} = \frac{1}{2} \frac{\partial}{\partial t}$ and $\mathcal{G} = \frac{1}{2} F'$. As a result, we can express the coefficients $u_{n,m}$ in terms of the parameters $\tau_k$ appearing in that transseries. The result is shown in table 4.

When we then plug in the values of $r_k$ given in (118), we obtain (cf. section 4.2 of [23])

$$\tau_k = 2k(-1)^k r_k = -\frac{(\mp \mathrm{i})^{k+1} \epsilon^k}{\pi}. \tag{122}$$

Remarkably, the parameters $\tau_k$ play two *distinct* roles: The parameters with odd $k$ depend on $\epsilon$ and encode the parity of the corresponding energy solution, whereas only the parameters with even $k$ behave as genuine transseries parameters implementing the Stokes phenomenon. This suggests that, if we want to write down a generic *one-parameter* transseries for the full quantum prepotential, only the $\tau_{2k}$ – or equivalently the $r_{2k}$ – can depend on the transseries parameter $\sigma$. However, as one can see from table 4, this implies that the coefficients $u_{n,m}$ are not homogeneous in the transseries parameter $\sigma$, and this prevents us from writing down a simple and elegant generic one-parameter transseries structure as we did in (94) and (110) for the cubic oscillator. We will come back to this issue in the next subsection.

Next, let us derive the Stokes data of this full quantum prepotential transseries. Analogous to (91), we define the building blocks of our energy transseries as

$$E_{n,m} = \left( \hbar \frac{\partial}{\partial t} \right)^m \left( \frac{\partial E}{\partial t} \lambda^{\frac{1}{2}n} \right). \tag{123}$$

Let $\tilde{\mathcal{A}} = \frac{1}{2} t_{D,0}$ denote the 'primitive' instanton action of the transseries (117), or equivalently of (121). Then an almost identical derivation to the one we performed for the cubic potential yields the alien calculus relations

$$\dot{\Delta}_{l\tilde{\mathcal{A}}} E_{n,m} = \frac{\tilde{S}_{l\tilde{\mathcal{A}}}(-1)^l}{2l} E_{n+l,m+1}, \tag{124}$$

which differ from (92) only by a factor $\frac{1}{2}$. The Stokes constants $\tilde{S}_{l\mathcal{A}}$ in this expression once again originate from relation (46), this time associated to the *full* quantum prepotential transseries where $\mathcal{G} = \frac{1}{2} t_D(t; \hbar)$. We can then match the action of the pointed alien derivative with the observed Stokes phenomenon in table 3. Because the first instanton correction is unambiguous, it does not 'jump', and so we have

$$\tilde{S}_{\tilde{\mathcal{A}}} = 0. \tag{125}$$

Table 4: The coefficients $u_{n,m}$ expressed in terms of the parameters $\tau_k$ of the full quantum prepotential transseries, which follows from considering (44) with $D = \frac{1}{2}\frac{\partial}{\partial t}$ and $\mathcal{G} = \frac{1}{2}F'$.

| $m$ | 0 | 1 | 2 | 3 | 4 |
|---|---|---|---|---|---|
| $u_{1,m}$ | $-\frac{\tau_1}{2}$ | | | | |
| $u_{2,m}$ | $\frac{\tau_2}{4}$ | $\frac{\tau_1^2}{8}$ | | | |
| $u_{3,m}$ | $-\frac{\tau_3}{6}$ | $-\frac{\tau_2\tau_1}{8}$ | $-\frac{\tau_1^3}{48}$ | | |
| $u_{4,m}$ | $\frac{\tau_4}{8}$ | $\frac{\tau_3\tau_1}{12} + \frac{\tau_2^2}{32}$ | $\frac{\tau_2\tau_1^2}{32}$ | $\frac{\tau_1^4}{384}$ | |
| $u_{5,m}$ | $-\frac{\tau_5}{10}$ | $-\frac{\tau_4\tau_1}{16} - \frac{\tau_3\tau_2}{24}$ | $-\frac{\tau_3\tau_1^2}{48} - \frac{\tau_2^2\tau_1}{64}$ | $-\frac{\tau_2\tau_1^3}{192}$ | $-\frac{\tau_1^5}{3840}$ |

At the two-instanton level we observe a jump in the coefficient $u_{2,0}$, which must be the consequence of $\dot{\Delta}_{2\tilde{\mathcal{A}}}$ acting on the perturbative energy $E$, which we can think of (see also figure 6) as $E_{0,-1}$. This implies

$$\tilde{S}_{2\tilde{\mathcal{A}}} = -\frac{2}{\mathrm{i}\pi}\,. \tag{126}$$

By extending this procedure to the higher orders we are led to infer that

$$\tilde{S}_{l\tilde{\mathcal{A}}} = \begin{cases} 0\,, & l \text{ odd}, \\ \frac{2(-1)^{l/2}}{\pi \mathrm{i}}\,, & l \text{ even}. \end{cases} \tag{127}$$

At first sight, this seems to be at odds with results [24] obtained using numerical methods, but the reason for this apparent discrepancy is that the we are working with the 'wrong' normalisation $\tilde{\mathcal{A}} = \frac{1}{2}t_{D,0}(E)$. In order to derive the Stokes data in the 'correct' normalisation, we need to consider the minimal transseries, which we demonstrate in the next subsection.

Let us end the discussion of the full transseries by pointing out an elegant feature it has. One could wonder what happens to the transseries for the energy (117) or the (full) quantum prepotential (121) if we take the unquantised variable $t$ and rotate it once around the origin $t = 0$ in the complex $t$-plane. Performing such a rotation, we find that the classical dual period $t_{D,0}(t)$ obtains a shift:

$$\frac{1}{2}t_{D,0}(t) \to \frac{1}{2}t_{D,0}(\mathrm{e}^{2\pi\mathrm{i}}\,t) = \frac{1}{2}t_{D,0}(t) + 2\pi\mathrm{i}\,t\,, \tag{128}$$

whereas its quantum corrections are invariant. For our transseries, this means that the non-perturbative sectors effectively obtain an additional factor:

$$\tilde{F}^{(n)} \to \tilde{F}^{(n)}\,\mathrm{e}^{2\pi\mathrm{i}nt/\hbar}\,, \quad \text{or equivalently} \quad E^{(n)} \to E^{(n)}\,\mathrm{e}^{2\pi\mathrm{i}nt/\hbar}\,. \tag{129}$$

It is then after quantising $t$ that all instanton sectors get a factor $(-1)^n$, and so the monodromy in the $t$-plane effectively swaps parity.[21] In terms of parameters $\tau_k$, this action swaps the sign

---

[21]Note that for the cubic oscillator, the instanton corrections have $\mathcal{A} = t_{D,0}(t)$ as the 'primitive' action, and therefore its energy transseries is invariant under a full rotation of $t$.

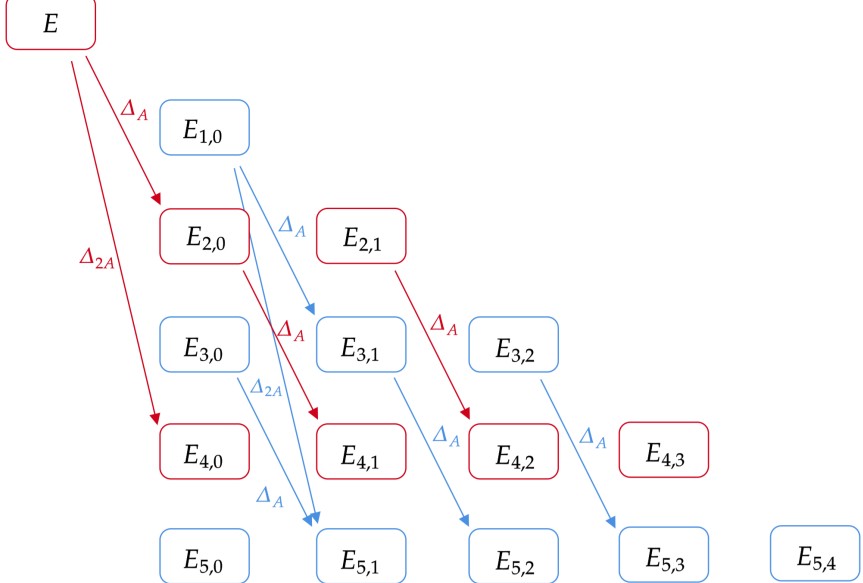

Figure 6: The alien lattice of the ordinary energy transseries of the double well potential as follows from (132). The *even* sectors and any Stokes derivatives connecting them are red, whereas all *odd* sectors and their alien derivatives are blue.

of all odd-$k$ parameters. From a more mathematical point of view, we therefore arrive at the conclusion that there are two types of actions (or automorphisms) on the space of transseries spanned by the parameters $\tau_k$: A Stokes phenomenon in the $\hbar$-plane which affects the even transseries parameters, and a monodromy in the $t$-plane which affects the odd transseries parameters.

## 4.2 The minimal transseries

We now turn to the *minimal* quantum prepotential transseries $\hat{F}^{\min}$. This transseries contains only sectors associated to singularities that appear in the Borel plane of its perturbative series, and therefore a natural Ansatz is

$$\hat{F}^{\min} = \sum_{n=0}^{\infty} F^{(n)}, \quad \text{with} \quad F^{(n)} \sim \lambda^n. \tag{130}$$

Note how the $n$-th instanton correction $F^{(n)}$ to the quantum prepotential $\hat{F}^{\min}$ runs parallel with the $2n$-th instanton correction $\tilde{F}^{(2n)}$ of the full transseries $\hat{F}^{\text{full}}$ – but be aware that they are not equal, as we will see shortly. Let us stick to the definition (123) for the building blocks $E_{n,m}$ of the energy transseries – where in particular, $n$ still denotes the power of $\lambda^{1/2}$. Under our Ansatz for the minimal quantum prepotential transseries, our alien calculus with $\mathcal{A} = t_{D,0}$ now leads to

$$\dot{\Delta}_{l\mathcal{A}} E = \frac{S_{l\mathcal{A}}(-1)^l}{l} \frac{\partial E}{\partial t} \lambda^l, \tag{131}$$

where the Stokes constants $S_{l\mathcal{A}}$ are once again defined by (46) and are associated to the minimal quantum prepotential transseries. This is the same expression as the one we found

Table 5: On the left the coefficients $u_{n,m}$ that appear in the *minimal* quantum pre-potential transseries. On the right those same coefficients expressed in terms of the parameters $\tau_k$ of the *full* quantum prepotential transseries.

| $m$ | 0 | 1 | 2 | ... | 0 | 1 | 2 | ... |
|---|---|---|---|---|---|---|---|---|
| $u_{1,m}$ | 0 | | | | 0 | | | |
| $u_{2,m}$ | $\mp\frac{\mathrm{i}}{4\pi}$ | 0 | | | $\frac{\tau_2}{4}$ | 0 | | |
| $u_{3,m}$ | 0 | 0 | 0 | | 0 | 0 | 0 | |
| $u_{4,m}$ | $\mp\frac{\mathrm{i}}{8\pi}$ | $-\frac{1}{32\pi^2}$ | 0 | ... | $\frac{\tau_4}{8}$ | $\frac{\tau_2^2}{32}$ | 0 | ... |
| $u_{5,m}$ | 0 | 0 | 0 | ... | 0 | 0 | 0 | ... |
| $u_{6,m}$ | $\mp\frac{\mathrm{i}}{12\pi}$ | $\frac{1}{32\pi^2}$ | $\pm\frac{\mathrm{i}}{384\pi^3}$ | ... | $\frac{\tau_6}{12}$ | $\frac{\tau_4\tau_2}{32}$ | $\frac{\tau_2^3}{384}$ | ... |

for the cubic in (90) and it generalises straightforwardly to

$$\dot{\Delta}_{l\mathcal{A}}E_{n,m} = \frac{S_{l\mathcal{A}}(-1)^l}{l}E_{n+2l,m+1}\,. \tag{132}$$

Comparing the constants $S_{l\mathcal{A}}$ with the ambiguities in table 3 then leads to the following value for the Stokes constants:

$$S_{l\mathcal{A}} = -\frac{1}{2\pi\mathrm{i}}\,, \tag{133}$$

which are the expected values, now matching the numerical results of [24]. Note that the Stokes constants, which are defined in relation (46), depend on the normalisation of the instanton sectors which itself is implicitly determined by the labelling of the sectors. From the factor $(-1)^{l+1}/l^2$ in (127) we can see how replacing $\tilde{\mathcal{A}}$ by $\mathcal{A} = 2\tilde{\mathcal{A}}$ changes the Stokes constants. In the large order analysis of [24], the first singularity in the Borel plane of the perturbative series is naturally identified with the first instanton correction, and hence the Stokes constants of the minimal quantum prepotential transseries given above emerge in their computation.

Now that we have obtained the Stokes data of the minimal transseries $\hat{F}^{\min}$, we would like to construct the minimal transseries itself. In order to do so, we recall equation (116), whose solution $\Delta t$ allows for the construction of the *full* transseries.[22] This equation can be separated into even and odd powers of the nonperturbative transmonomial $\lambda^{1/2}$, and summing those contributions into separate terms yields

$$\Delta t = \left(\pm\frac{1}{4\pi\mathrm{i}}\right)\underbrace{\hbar\log\left(1 + \mathrm{e}^{-\frac{1}{\hbar}t_D(t+\Delta t;\hbar)}\right)}_{\Delta t_{\mathrm{DDP}}} + \underbrace{\frac{\hbar}{2\pi}\arctan\left(-\epsilon\,\mathrm{e}^{-\frac{1}{2\hbar}t_D(t+\Delta t;\hbar)}\right)}_{\Delta t_{\mathrm{med}}(\epsilon)}\,, \tag{134}$$

where we used $(\pm)^2 = 1$. This expression is extremely useful because it splits the nonperturbative correction $\Delta t$ coming from the quantisaton condition into a 'resurgence' piece $\Delta t_{\mathrm{DDP}}$

---

[22]At this point we are going to allow ourselves to be sloppy with the language: We will speak of full and minimal transseries *without* specifically refering to $E$ or $F$, since both quantities are similar and closely related due to the PNP relation.

and a parity-dependent piece $\Delta t_{\mathrm{med}}(\epsilon)$. The former piece contains only even powers of the instanton transmonomial and therefore determines the values of all the $r_{2k}$ (or equivalently $\tau_{2k}$) in (119), which are solely responsible for the Stokes phenomenon – note also the similarity to (60) for the cubic. The latter piece, $\Delta t_{\mathrm{med}}(\epsilon)$, where 'med' stands for *median*, is responsible for all the $r_{2k-1}$ (or $\tau_{2k-1}$) and is real and unambiguous, but parity dependent. Its name and role will be explained in a moment.

Let us for the moment focus on the first piece and consider the new equation

$$\Delta t = \left(\pm \frac{1}{4\pi \mathrm{i}}\right)\Delta t_{\mathrm{DDP}} = \left(\pm \frac{1}{4\pi \mathrm{i}}\right)\hbar \log\left(1 + \mathrm{e}^{-\frac{1}{\hbar}t_D(t+\Delta t;\hbar)}\right). \tag{135}$$

We use the approach of section 3.5 again for this 'partial' $\Delta t$: consider $\varphi(\Delta t) = F'(t+\Delta t;\hbar)$, calculate the transseries extension for the quantum dual period $F'(t;\hbar)$ and integrate the result with respect to $t$. This yields a transseries that we call $\hat{F}^{\min}$ – as we will justify shortly – and which only contains integer powers of $\lambda$. The result is

$$\hat{F}^{\min} = F + \hbar^2 \sum_{n=1}^{\infty}\left[-\frac{2}{n}u_{n,0}\lambda^{\frac{1}{2}n} + \sum_{m=1}^{n-1} u_{n,m}\left(\hbar\frac{\partial}{\partial t}\right)^{m-1} F'' \lambda^{\frac{1}{2}n}\right], \tag{136}$$

where the $u_{n,m}$ receive as input data

$$r_{2k} = \frac{(-1)^k}{k}\left(\mp \frac{1}{4\pi \mathrm{i}}\right), \tag{137}$$
$$r_{2k-1} = 0,$$

for positive integers $k$. In table 5 we have displayed both the coefficients $u_{n,m}$ and $\tau_k$ that appear in this transseries; note in particular that all $u_{n,m}$ with odd $n$ vanish, so that indeed only integer powers of $\lambda$ survive in the minimal transseries. We want to argue that this transseries is in fact *the* minimal transseries associated to the perturbative quantum prepotential $F$: First of all note that all the sectors in the transseries are closed under the action of the alien derivatives as described by equation (132). Secondly, one can check that the ambiguities in table 5 are consistent with the Stokes constants (133). For example, from (132) we get $\dot{\Delta}_{2\mathcal{A}}E = \frac{\mathrm{i}}{4\pi}E_{2,0}$, which explains the observed ambiguity for $u_{2,0}$ in table 5.

The minimal transseries admits the same generic one-parameter transseries structure in terms of a single parameter $\sigma$ as we found for the cubic:

$$\hat{F}^{\min}(t,\sigma;\hbar) = F + \hbar^2 \sum_{n=1}^{\infty}\left[-\frac{1}{n}u_{2n,0}\,\sigma\lambda^n + \sum_{m=1}^{n-1} u_{2n,m}\sigma^{m+1}\left(\hbar\frac{\partial}{\partial t}\right)^{m-1} F'' \lambda^n\right], \tag{138}$$

where now $u_{2n,m}$ is computed using $r_{2k} = \frac{(-1)^k}{k}$. Analogously, for the same values $r_{2k}$, there is a 'minimal energy transseries'

$$\hat{E}^{\min}(t,\sigma;\hbar) = E + \hbar \sum_{n=1}^{\infty}\sum_{m=0}^{n-1} u_{2n,m}\sigma^{m+1}\left(\hbar\frac{\partial}{\partial t}\right)^{m} E' \lambda^n, \tag{139}$$

which has a form identical to that of the one-parameter transseries (94) that we found in the cubic model. We can then check explicitly that

$$\mathfrak{S}_0 E(t;\hbar) = \hat{E}^{\min}\left(t, -\frac{1}{2\pi \mathrm{i}};\hbar\right), \tag{140}$$

and hence (139) truly is the minimal transseries extension associated to the perturbative energy. In the same way, we can verify that

$$\mathfrak{S}_0 F(t;\hbar) = \hat{F}^{\min}\left(t, -\frac{1}{2\pi \mathrm{i}};\hbar\right). \tag{141}$$

Furthermore, table 5 simply depicts the coefficients $u_{n,m}\sigma^{m+1}$ of the transseries $\hat{E}^{\min}(t,\pm\frac{1}{4\pi i};\hbar)$.

Both transseries (138) and (139) can also be generated using our algorithm by redefining $\Delta t$ in equation (135) as

$$\Delta t(\sigma) = \sigma\,\Delta t_{\mathrm{DDP}} = \sigma\,\hbar\log\left(1 + e^{-\frac{1}{\hbar}t_D(t+\Delta t(\sigma);\hbar)}\right),\qquad(142)$$

which allows us to produce the generic one-parameter transseries in one fell swoop. For example, we have

$$\hat{E}^{\min}(t,\sigma;\hbar) = E(t + \Delta t(\sigma);\hbar).\qquad(143)$$

for the one-parameter energy transseries.

## 4.3 The full transseries as a sum of minimal transseries

The minimal transseries discussed in the previous subsection contains only a subset of all nonperturbative sectors of the full energy or quantum prepotential transseries that we are interested in and that we discussed in section 4.1. However, we can write the full transseries as a *sum* of minimal transseries, and when we do so an interesting structure emerges: The full transseries 'factorises' in terms of a minimal transseries and another transseries that we will call 'median transseries'.

To arrive at this conclusion, we first observe that each sector $E_{n,m}$ gives rise to a minimal transseries extension of itself:

$$\mathfrak{S}_0 E_{n,m} = E_{n,m} - S_{\mathcal{A}}E_{n+2,m+1} + \left(\frac{1}{2}S_{2\mathcal{A}}E_{n+4,m+1} + \frac{1}{2}S_{\mathcal{A}}^2 E_{n+4,m+2}\right) + \dots\qquad(144)$$

We would like to argue that our algorithm with $\Delta t(\sigma)$ from (142) can produce the generic one-parameter minimal transseries which describes this Stokes phenomenon. To this end, let us define the transseries

$$\hat{E}_{n,m}^{\min}(t,\sigma;\hbar) = E_{n,m}(t + \Delta t(\sigma);\hbar).\qquad(145)$$

Then, given definition (123), we also have

$$\hat{E}_{n,m}^{\min}(t,\sigma;\hbar) = (\hbar\partial_t)^m\left(\partial_t\hat{E}^{\min}(t,\sigma;\hbar)e^{-\frac{n}{2\hbar}\partial_t\hat{F}^{\min}(t,\sigma;\hbar)}\right).\qquad(146)$$

When we expand the right hand side of this equation using (138) and (139) in powers of the transmonomial $\lambda$ and regroup the resulting expression in terms of building blocks (123), we find for the first few orders that

$$\hat{E}_{n,m}^{\min}(t,\sigma;\hbar) = E_{n,m} - \sigma E_{n+2,m+1} + \left(\frac{1}{2}\sigma E_{n+4,m+1} + \frac{1}{2}\sigma^2 E_{n+4,m+2}\right) + \dots\qquad(147)$$

As advocated, this captures the action of the Stokes automorphism in (144):

$$\mathfrak{S}_0\hat{E}_{n,m}^{\min}(t,0;\hbar) = \hat{E}_{n,m}^{\min}(t,S_{\mathcal{A}};\hbar),\qquad(148)$$

where we use the fact (133) that all $S_{l\mathcal{A}}$ are equal. The minimal transseries extension (147) of generic nonperturbative sectors can actually be written in closed form as

$$\hat{E}_{n,m}^{\min}(t,\sigma;\hbar) = E_{n,m} + \sum_{n'=1}^{\infty}\sum_{m'=0}^{n'-1} u_{n',m'}\sigma^{m'+1}E_{n+n',m+m'+1},\qquad(149)$$

with the $u_{n',m'}$ defined by $r_{2k} = \frac{(-1)^k}{k}$ and $r_{2k-1} = 0$. This expression also covers the original minimal transseries $E^{\min}(\sigma)$ given in (139) if we identify $E_{0,-1}$ with the perturbative energy $E$.

Note how this is indeed the natural identification from the alien calculus point of view – see for example figure 6.

To retrieve the full transseries as a sum of minimal transseries, we then need to know which minimal transseries $\hat{E}_{n,m}^{\min}$ we need to sum. This is where the second piece of the right hand side of equation (134) comes into play: let us define

$$\Delta t(\epsilon) = \Delta t_{\mathrm{med}}(\epsilon) = \frac{\hbar}{2\pi} \arctan\left(-\epsilon\, \mathrm{e}^{-\frac{1}{2\hbar} t_D(t+\Delta t(\epsilon);\hbar)}\right). \tag{150}$$

Then, following the by now familiar procedure outlined in section 3.5, with $\varphi(\Delta t(\epsilon)) = E(t + \Delta t(\epsilon);\hbar)$, we obtain:

$$\hat{E}_{\epsilon}^{\mathrm{med}}(t;\hbar) = E + \hbar \sum_{n=1}^{\infty} \sum_{m=0}^{n-1} u_{2n,m}(\epsilon) \left(\hbar \frac{\partial}{\partial t}\right)^m E' \lambda^{n/2}, \tag{151}$$

with input data

$$r_{2k} = 0,$$
$$r_{2k-1} = \frac{\epsilon}{2\pi} \frac{(-1)^k}{2k-1}, \tag{152}$$

for positive integers $k$, which follows from comparing (150) to (119). The coefficients $u_{n,m}$ for this 'median' transseries are displayed in table 6. Note that this transseries is completely *real* and *unambiguous* and its median resummation (see e.g. [17, 49]) provides the easiest way to obtain the real energy spectrum of the physical quartic oscillator to numerically high precision.[23] We elaborate on this statement in appendix C.

Now we are ready to formulate the full transseries as a sum of minimal transseries. First, we note that the *true* equation for $\Delta t$ given in (134) can be extended to the *two-parameter* generalisation

$$\Delta t(\sigma,\epsilon) = \sigma \Delta t_{\mathrm{DDP}} + \Delta t_{\mathrm{med}}(\epsilon) \tag{153}$$
$$= \sigma \hbar \log\left(1 + \mathrm{e}^{-\frac{1}{\hbar} t_D(t+\Delta t(\sigma,\epsilon);\hbar)}\right) + \frac{\hbar}{2\pi} \arctan\left(-\epsilon\, \mathrm{e}^{-\frac{1}{2\hbar} t_D(t+\Delta t(\sigma,\epsilon);\hbar)}\right).$$

We can retrieve (134) by setting $\sigma = \mp \frac{1}{4\pi \mathrm{i}}$ in the above definition. We can then formulate a *two-parameter transseries* for the full energy transseries:

$$\hat{E}_{\epsilon}^{\mathrm{full}}(t,\sigma;\hbar) = \hat{E}_{\epsilon}^{\mathrm{med}}(t + \sigma \Delta t_{\mathrm{DDP}};\hbar)$$
$$= \hat{E}^{\min}(\sigma) + \hbar \sum_{n=1}^{\infty} \sum_{m=0}^{n-1} u_{n,m}(\epsilon) \left(\hbar \frac{\partial}{\partial t}\right)^m \partial_t \hat{E}^{\min}(\sigma) \mathrm{e}^{-\frac{n}{2\hbar} \partial_t \hat{F}^{\min}(\sigma)}$$
$$= \hat{E}^{\min}(\sigma) + \hbar \sum_{n=1}^{\infty} \sum_{m=0}^{n-1} u_{n,m}(\epsilon) \hat{E}_{n,m}^{\min}(\sigma), \tag{154}$$

with the $r_k$ defined in (152). This is the sum of minimal transseries that we advocated at the start of this subsection. Upon close inspection, one can recognize this structure in table 4: With every *unique* product of parameters $\tau_{2k-1}$ with odd indices we can identify a minimal resurgent transseries $E_{n,m}^{\min}$ in the sum (154). For example, the minimal resurgent transseries $\hat{E}_{1,0}^{\min}$ is based on $\tau_1$, and is constructed from those sectors that carry factors $\{\tau_1, \tau_1\tau_2, \tau_1\tau_4, \tau_1\tau_2^2,\ldots\}$, where $\tau_1$ is only multiplied by further $\tau_{2k}$ with *even* indices. The corresponding sectors are $\{E_{1,0}, E_{3,1}, E_{5,1}, E_{5,2},\ldots\}$ respectively. Note that conversely, this construction also allows us to

---

[23]Alternatively, one can consider *exact perturbation theory* [50, 51] which approximates these physical values without the use of transseries altogether.

write the full energy transseries as a sum of median transseries, weighted by the $u_{n,m}$ of the minimal transeries (137) – since we can identify a median transseries with a unique product of even $\tau_{2k}$. Heuristically, we have

$$\text{Full transseries} \simeq \text{Minimal transseries} \otimes \text{Median transseries.}$$

That the full transseries can be written as a sum of minimal transseries was to be expected: if a transseries is not minimal, one should at least be able to partition it into separate parts that are minimal. What is striking however is that it truly *factorises*, in the sense that *all* minimal transseries extensions of the sectors in the median transseries have the same structure (149). The full transseries is really a 'product' of both structures, which can be best understood from the split form of the shift $\Delta t(\sigma, \epsilon)$ in (153).

We conclude this section by highlighting some properties of our two-parameter transseries. First of all, let $E_{\epsilon,N}$ denote the physical values of the nonperturbative energy levels of the symmetric double well potential. These are obtained from our two-parameter transseries in the following way:

$$E_{\epsilon,N}(\hbar) = \mathcal{S}_{0^{\pm}}\hat{E}_{\epsilon}^{\text{full}}\left(\hbar\left(N + \frac{1}{2}\right), \pm\frac{1}{4\pi\mathrm{i}}; \hbar\right). \tag{155}$$

Note that the answer depends on the quantum numbers $(\epsilon, N)$, but is independent of the resummation prescription $\pm$, as long as we pick the corresponding value for the transseries parameter. In the language of resurgence, the Stokes phenomenon that we observe along the Stokes ray $\arg(\hbar) = 0$, and which takes us from the $D_{\epsilon}^{+} = 0$ to the $D_{\epsilon}^{-} = 0$ quantisation condition (113), is captured by

$$\mathfrak{S}_0\hat{E}_{\epsilon}^{\text{full}}\left(t, \frac{1}{4\pi\mathrm{i}}; \hbar\right) = \hat{E}_{\epsilon}^{\text{full}}\left(t, -\frac{1}{4\pi\mathrm{i}}; \hbar\right). \tag{156}$$

The monodromy of the transseries under the rotation $t \mapsto t\,\mathrm{e}^{2\pi\mathrm{i}}$ that we discussed earlier also has a clear manifestation in the two-parameter transseries setting: it acts as

$$\hat{E}_{\epsilon}^{\text{full}}(t, \sigma; \hbar) \to \hat{E}_{-\epsilon}^{\text{full}}(t, \sigma; \hbar). \tag{157}$$

The last property we want to mention is that the *minimal* and *median* transseries are easily obtained from the full two-parameter transseries when we set[24] $\epsilon$ or $\sigma$ to zero respectively:

$$\hat{E}_{0}^{\text{full}}(t, \sigma; \hbar) = \hat{E}^{\text{min}}(t, \sigma; \hbar),$$
$$\hat{E}_{\epsilon}^{\text{full}}(t, 0; \hbar) = \hat{E}_{\epsilon}^{\text{med}}(t; \hbar). \tag{158}$$

Of course, all the properties we have mentioned here for the energy transseries can similarly be deduced for a full two-parameter quantum prepotential transseries $\hat{F}_{\epsilon}^{\text{full}}(t, \sigma; \hbar)$.

## 5 The cosine potential

The last example that we want to discuss is the periodic cosine potential

$$V(x) = 1 - \cos(x), \tag{159}$$

which in the literature is also sometimes called the sine-Gordon potential and its Schrödinger equation the Mathieu equation. We have two reasons for studying this potential: first, to

---

[24]Note that physcially $\epsilon$ cannot be set to zero, so this is only mathematical trick to illustrate the factorisation of the full transseries in terms of the minimal and median transseries.

Table 6: Values for the coefficients $u_{n,m}$ in the median energy transseries of the double well potential.

| $m$ | 0 | 1 | 2 | 3 | 4 |
|---|---|---|---|---|---|
| $u_{1,m}$ | $-\frac{1}{2\pi}\epsilon$ | | | | |
| $u_{2,m}$ | 0 | $\frac{1}{8\pi^2}$ | | | |
| $u_{3,m}$ | $\frac{1}{6\pi}\epsilon$ | 0 | $-\frac{1}{48\pi^3}\epsilon$ | | |
| $u_{4,m}$ | 0 | $-\frac{1}{12\pi^2}$ | 0 | $\frac{1}{384\pi^4}$ | |
| $u_{5,m}$ | $-\frac{1}{10\pi}\epsilon$ | 0 | $\frac{1}{48\pi^3}\epsilon$ | 0 | $-\frac{1}{3840\pi^5}\epsilon$ |

demonstrate that the intricate transseries structure of the energy of the double well potential occurs more generally and also applies to the cosine potential. Secondly, in the cosine potential the energy eigenstates are labeled by a continous periodic parameter $\theta$ that allows us to identify and compare *topological sectors* to *transseries sectors*.

We take $0 < E < 2$ so that the turning points are real, and start by formulating the exact quantisation condition for the potential, which can be computed using the aforementioned techniques and which can be found in [14, 27, 52]. To keep the notation tidy, we express the quantisation conditions in terms of Voros symbols (27):

$$D_\theta^\pm = 1 + \mathcal{V}_A^{\mp 1}(1 + \mathcal{V}_B) - 2\sqrt{\mathcal{V}_A^{\mp 1}\mathcal{V}_B}\cos(\theta) = 0\,, \tag{160}$$

where $\mathcal{V}_A$ and $\mathcal{V}_B$ are the Voros symbols associated to the perturbative and tunneling cycles respectively. As with the double well, we observe a *single* tunneling contribution from the fact that there is a power of $\frac{1}{2}$ in the $\sqrt{\mathcal{V}_B}$ in the above equation, hinting at a similar full transseries structure to the one we found for the double well potential. There is however also a novelty here: Compared to the double well potential, for which the basis states are labeled by the parity $\epsilon$ which takes on two values – and which relates to the $\mathbb{Z}_2$ symmetry of the potential – we now have states labeled by the so-called *Bloch angle* $\theta$ which takes any value between 0 and $2\pi$ – and which relates to the $\mathbb{Z}$ translational symmetry of the potential. The Bloch angle originates from the *Bloch boundary condition* on the wave functions,

$$\psi(x + 2\pi) = e^{i\theta}\psi(x)\,, \tag{161}$$

which gives rise to the quantisation condition above – see e.g. [52]. At the perturbative level – that is, ignoring as before all factors of $\mathcal{V}_B$ – the quantisation condition becomes $\theta$-independent and reduces to

$$1 + \mathcal{V}_A^{\mp 1} = 0\,, \quad \text{implying} \quad t = \hbar\left(N + \frac{1}{2}\right)\,. \tag{162}$$

Subsequently, promoting $t$ to $\hat{t} = t + \Delta t$ in the same manner as before, we can reduce the

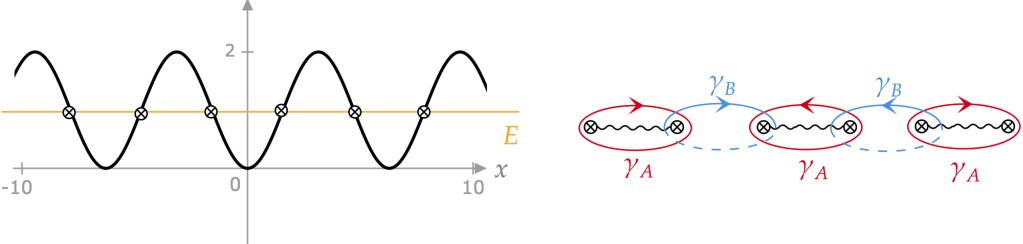

Figure 7: On the left we draw the cosine potential (159), and on the right some of its infinite number of turning points in the complex $x$-plane, with the cycles $\gamma_A$ and $\gamma_B$.

exact quantisation condition[25] to an equation for $\Delta t$. This leads to

$$\Delta t = \mp \frac{\mathrm{i}\hbar}{\pi} \log\left( \pm\mathrm{i}\cos(\theta)\,\mathrm{e}^{-t_D(t+\Delta t;\hbar)/2\hbar} + \sqrt{1 + \sin^2(\theta)\,\mathrm{e}^{-t_D(t+\Delta t;\hbar)/\hbar}} \right). \qquad (164)$$

We then solve the above equation by using our algorithm and produce a transseries for the energy. Note that in the equation above, we again have an instanton transmonomial with $\frac{1}{2}t_D$ in the exponent, and hence the correct Ansatz for the full energy transseries is

$$\hat{E} = E + \hbar \sum_{n=1}^{\infty}\sum_{m=0}^{n-1} u_{n,m}\left(\hbar\frac{\partial}{\partial t}\right)^m \left(\frac{\partial E}{\partial t}\lambda^{\frac{1}{2}n}\right), \qquad (165)$$

which contains an expansion in $\lambda^{\frac{1}{2}}$. Our algorithm computes the coefficients $u_{n,m}$ which are tabulated in table 7.

A few comments are in order. At the nonperturbative level, the energy levels spread out into continuous *bands* that are doubly degenerate ($E_\theta = E_{-\theta}$), contrary to the double well, whose energy levels are nondegenerate at the nonperturbative level. Every energy level in a band is characterised by its Bloch angle $\theta$, and the edges of the bands correspond to $\theta = 0$ (upper bound) and $\theta = \pi$ (lower bound). When we set $\theta = \frac{1}{2}\pi$ or $\theta = \frac{3}{2}\pi$, we retrieve the minimal transseries, as we will explain shortly. Another interesting feature is the rotation of the period $t$ around the origin in the complex plane, which produces a factor $(-1)^n$ for the instanton sector $E^{(n)}$ after quantisation (162). Given the structure of the energy transseries (table 7), one can easily see that effectively, this monodromy can be effectuated by sending $\theta \mapsto \theta + \pi$.

As explained in [15], the Bloch angle $\theta$ can be incorporated into a path integral description of the problem by adding a topological term to the action. From this perspective, the $\theta$-dependence of the nonperturbative corrections indicates which saddle point solution is responsible for a particular nonperturbative correction. When a solution tunnels to the right (left) it obtains a factor $\mathrm{e}^{\mathrm{i}\theta}$ ($\mathrm{e}^{-\mathrm{i}\theta}$) and we can assign it a charge $+1$ ($-1$). These solutions are often called (anti-) instantons in the literature. We can then assign a *total* charge to every saddle point solution which is a composition of instantons and anti-instantons, and in this way distinguish different *topological sectors* of solutions. It naturally follows that resurgence can only mix transseries sectors in the same topological sector, as resurgence analysis of $\hbar$-expansions is completely independent of $\theta$. This notion is often graphically depicted in what

---

[25] In doing so one must be careful with the branch of the square root:

$$\sqrt{\mathcal{V}_A^{\pm 1}} = \pm\mathrm{i}\mathrm{e}^{\pm\pi\mathrm{i}\Delta t/\hbar}. \qquad (163)$$

In principle, one should also have a factor of $(-1)^N$ on the right hand side of this expression, but we can absorb this into $\cos\theta$ by a suitable redefinition of $\theta$.

is called the resurgence triangle [26, 27]. In this section we wish to understand the relation between topological sectors and transseries sectors.

Let us take a closer look at $\Delta t$, and at the $r_k$ that it produces in equation (119). We can again split $\Delta t$ into two parts: a resurgence part $\Delta t_{\text{DDP}}$ and median part $\Delta t_{\text{med}}$, which generate the even $r_{2k}$ and odd $r_{2k-1}$ respectively. For the cosine potential, this decomposition reads

$$\Delta t = \left(\pm \frac{1}{2\pi i}\right) \underbrace{\hbar \log\left(1 + e^{-\frac{1}{\hbar}t_D(t+\Delta t;\hbar)}\right)}_{\Delta t_{\text{DDP}}} + \underbrace{\frac{\hbar}{\pi} \arcsin\left(\cos(\theta)\frac{e^{-\frac{1}{2\hbar}t_D(t+\Delta t;\hbar)}}{\sqrt{1 + e^{-\frac{1}{\hbar}t_D(t+\Delta t;\hbar)}}}\right)}_{\Delta t_{\text{med}}}. \quad (166)$$

Remarkably, the form of equation (164) is such that in the expansion all contributions conspire to cancel the $\theta$ dependence of the $r_{2k}$, and hence the first term on the right hand side of the above equation is also independent of the angle. From the point of view developed in the previous section, this is exactly as one would expect: the $r_{2k}$ should parametrize the Stokes phenomenon, and hence we expect them to be ambiguous but independent of the Bloch angle $\theta$. In fact, the $\Delta t_{\text{DDP}}$ in the present context is identical to the one in (134) for the double well potential. The only subtle difference is that its prefactor is twice the prefactor in (134), which is a consequence of the fact that the Stokes constants $S_{lA}$ in the cosine potential are twice the size of the Stokes constants of the double well potential. Let us derive this fact: if we consider the *minimal* transseries extension of $E$, and its associated minimal quantum prepotential

$$F^{\text{min}} = \sum_{n=0}^{\infty} F^{(n)}, \quad \text{with} \quad F^{(n)} \sim \lambda^n, \quad (167)$$

then we can map the alien calculus (46) of the quantum prepotential back to the energy via the PNP relation. Following the by now familiar procedure, we obtain the same relations as for the double well, which read

$$\dot{\Delta}_{lA} E_{n,m} = \frac{S_{lA}(-1)^l}{l} E_{n+2l,m+1}, \quad (168)$$

where the $E_{n,m}$ are defined as in (123). Comparing this to the entries in table 7 leads to the Stokes constants

$$S_{lA} = -\frac{1}{\pi i}, \quad (169)$$

Table 7: Values of the coefficients $u_{n,m}$ for the full energy transseries of the cosine potential. The index $n$ labels the rows, $m$ labels the columns and $\Theta = \cos(\theta)$.

| $m$ | 0 | 1 | 2 | 3 | 4 |
|---|---|---|---|---|---|
| $E^{(1)}$ | $\frac{1}{\pi}\Theta$ | | | | |
| $E^{(2)}$ | $\mp\frac{i}{2\pi}$ | $\frac{1}{2\pi^2}\Theta^2$ | | | |
| $E^{(3)}$ | $-\frac{1}{6\pi}\left(\Theta^3 - 3\Theta\right)$ | $\mp\frac{i}{2\pi^2}\Theta$ | $\frac{1}{6\pi^3}\Theta^3$ | | |
| $E^{(4)}$ | $\pm\frac{i}{4\pi}$ | $\frac{1}{24\pi^2}\left(4\Theta^4 - 12\Theta^2 - 3\right)$ | $\mp\frac{i}{4\pi^3}\Theta^2$ | $\frac{1}{24\pi^4}\Theta^4$ | |
| $E^{(5)}$ | $\frac{1}{40\pi}\left(3\Theta^5 - 10\Theta^3 + 15\Theta\right)$ | $\mp\frac{1}{12\pi^2}\left(\Theta^3 - 6\Theta\right)$ | $\frac{1}{24\pi^3}\left(2\Theta^5 - 6\Theta^3 + 3\Theta\right)$ | $\mp\frac{i}{12\pi^4}\Theta^3$ | $\frac{1}{120\pi^5}\Theta^5$ |

Table 8: Values of the coefficients $u_{n,m}$ for the median energy transseries of the cosine potential. The index $n$ labels the rows, $m$ labels the columns, and $\Theta = \cos(\theta)$.

| $m$ | 0 | 1 | 2 | 3 | 4 |
|---|---|---|---|---|---|
| $u_{1,m}$ | $\frac{1}{\pi}\Theta$ | | | | |
| $u_{2,m}$ | 0 | $\frac{1}{2\pi^2}\Theta^2$ | | | |
| $u_{3,m}$ | $\frac{1}{6\pi}\left(\Theta^3 - 3\Theta\right)$ | 0 | $\frac{1}{6\pi^3}\Theta^3$ | | |
| $u_{4,m}$ | 0 | $\frac{1}{24\pi^2}\left(4\Theta^4 - 12\Theta^2\right)$ | 0 | $\frac{1}{24\pi^4}\Theta^4$ | |
| $u_{5,m}$ | $\frac{1}{40\pi}\left(3\Theta^5 - 10\Theta^3 + 15\Theta\right)$ | 0 | $\frac{1}{24\pi^3}\left(2\Theta^5 - 6\Theta^3\right)$ | 0 | $\frac{1}{120\pi^5}\Theta^5$ |

which indeed are twice the Stokes constants we found for the cubic (93) and double well potential (133). The geometrical reason for this factor of two is the fact that the A-cycles in the cosine potential intersect not one but *two* B-cycles – see figure 7.

Let us now focus on the median part of $\Delta t$, which is the second term on the right hand side of (166) and which generates all odd $r_{2k-1}$. If we restrict ourselves to

$$\Delta t(\theta) = \Delta t_{\text{med}}(\theta) = \frac{\hbar}{\pi}\arcsin\left(\cos(\theta)\,\frac{e^{-\frac{1}{2\hbar}t_D(t+\Delta t(\theta);\hbar)}}{\sqrt{1 + e^{-\frac{1}{\hbar}t_D(t+\Delta t(\theta);\hbar)}}}\right), \qquad (170)$$

then we find the coefficients $u_{n,m}$ in table 8 that define the median energy transseries for cosine potential. As with the double well, when we enhance these sectors in the median transseries to their respective minimal transseries extensions, we obtain the full transseries.

We now want to address the question of whether and how topological sectors relate to transseries sectors. Let us first remark that generally we cannot assign a single topological charge to a transseries sector: the coefficients $u_{n,m}$ are polynomials in $\cos(\theta) = \frac{1}{2}(e^{i\theta} + e^{-i\theta})$ and therefore we assign multiple topological charges (or sectors) to a single transseries sector. As we argued before, a necessary condition for resurgence to connect two transseries sectors is for these sectors to share a topological charge – i.e. they must share the same power of $e^{i\theta}$ in their $u_{n,m}$. The median transseries, however, indicates that this is *not* a sufficient condition: We can observe that several transseries sectors share similar powers of $e^{i\theta}$, yet are disconnected in a resurgence sense. For example, the two-instanton sector $E_{2,1}$ is fully invisible to the perturbative series $E$, yet it contains a constant part which has topological charge 0. Therefore, we can conclude that a single topological sector within the energy transseries is *not* connected from a resurgence perspective and moreover that the resurgence triangle does not partition the full transseries to single minimal transseries.

Curiously, the structure of the median energy transseries for the cosine potential is to a certain extent identical to that of the double well: their median transseries are constructed from the same nonzero coefficients $u_{n,m}$, even though the values of these coefficients are different. We thus observe that despite the different topological nature of both quantum systems (one potential having a quantum number coming from a $\mathbb{Z}_2$ reflection symmetry, the other having one coming from a $\mathbb{Z}$ translational symmetry), the resurgent structures of their respective transseries are qualitatively identical. For example, the alien lattice of the double well as graphically depicted in figure 6 describes the energy transseries of the cosine potential equally

well. The qualitative distinction does not lie in the transseries expression for an individual energy eigenstate. Rather – as also described in [52] for periodic potentials with $N$ minima on a circle – the topology affects the structure[26] of the basis of energy eigenstates for the Hilbert space.

Let us wrap up this discussion by generalising the above transseries to a two-parameter transseries. We first note that the minimal transseries expression for the cosine potential has the same universal form (139) as the double well (and cubic) potential, with $r_{2k} = \frac{(-1)^k}{k}$ and $r_{2k-1} = 0$. We then introduce

$$\Delta t(\sigma, \theta) = \sigma \Delta t_{\mathrm{DDP}} + \Delta t_{\mathrm{med}}(\theta), \tag{171}$$

and we write the full energy transseries as a sum of minimal transseries weighed by the $u_{n,m}$ of the median transseries. That is:

$$\begin{aligned}
\hat{E}_\theta^{\mathrm{full}}(t, \sigma; \hbar) &= E(t + \sigma \Delta t_{\mathrm{DDP}} + \Delta t_{\mathrm{med}}(\theta); \hbar) \\
&= \hat{E}^{\mathrm{min}}(\sigma) + \hbar \sum_{n=1}^{\infty} \sum_{m=0}^{n-1} u_{n,m} \hat{E}_{n,m}^{\mathrm{min}}(\sigma),
\end{aligned} \tag{172}$$

where the $\hat{E}_{n,m}^{\mathrm{min}}$ are the minimal transseries extensions of their respective transseries sectors as explained in the previous section. The physical energy levels that fill the energy bands of the cosine potential are then given by

$$E_{\theta,N} = \mathcal{S}_{0^\pm} \hat{E}_\theta^{\mathrm{full}}\left(\hbar\left(N + \frac{1}{2}\right), \pm\frac{1}{2\pi \mathrm{i}}; \hbar\right), \tag{173}$$

where $N$ is a nonnegative integer – see footnote 12. Finally, we note that the Stokes phenomenon that maps the quantisation condition $D^+ = 0$ to $D^- = 0$ corresponds to

$$\mathfrak{S}_0 \hat{E}_\theta^{\mathrm{full}}(t, +\frac{1}{2\pi \mathrm{i}}; \hbar) = \hat{E}_\theta^{\mathrm{full}}\left(t, -\frac{1}{2\pi \mathrm{i}}; \hbar\right), \tag{174}$$

and that the minimal transseries is obtained from setting $\theta = \frac{1}{2}\pi$ or $\theta = \frac{3}{2}\pi$ and the median transseries from setting $\sigma = 0$.

# 6 Conclusion

In this paper we have extended the computation of nonperturbative energy spectrum corrections for one-dimensional quantum oscillators to all orders, and unified them in a single transseries description. Using alien calculus, we have dissected the Stokes phenomenon and computed the asociated Stokes constants in two different ways. For the cubic oscillator we found that the transseries is minimal, and as a result it can be generalised to a one-parameter transseries which fully describes the Stokes phenomenon along the positive real $\hbar$-axis. For the double well and cosine potentials, we have nonperturbative corrections associated to instanton solutions that tunnel an odd number of times, and therefore the resulting transseries structure is no longer minimal. We have demonstrated that this more complicated transseries structure can be captured in terms of a two-parameter transseries which 'factorises' into a minimal and a median transseries.

One question left open by our work concerns the median transseries of sections 4 and 5, which consist of a countably infinite number of nonperturbative sectors that need to be

---

[26]See also [53] for a recent discussion on the similarities between oscillators with distinct vacuum structures.

resummed. The resurgence triangle, though helpful, does not conclusively explain this structure. In quantum field theory, for the $O(N)$ non-linear sigma model, it was shown [54] that the ambiguity-free transseries whose median resummation produces the physical value for the energy density – and which is the analogue of the median transseries described here – consists of a *finite*[27] number of sectors. It would be interesting to understand from a topological point of view why in this respect, one-dimensional quantum mechanics models seem more complicated than their quantum field theory cousins.

There are several other directions that we think deserve further investigation. For one, it would be interesting to extend our analysis of the Stokes phenomenon for the minimal quantum prepotential transseries to *all* directions in the complex $\hbar$-plane. In the quantum mechanical examples discussed here, within the parameter regime where all turning points are real, this would involve four Stokes rays corresponding to four primitive instanton actions $\mathcal{A} = \pm t_0$ and $\mathcal{A} = \pm t_{D,0}$, that we suspect would lead to a *four-parameter* transseries in the minimal case. One could then cover its complete alien calculus including the so-called 'backwards resurgence' [30] and inspect the action of the monodromy from rotation in the complex $t$-plane on this transseries.

Finally, another interesting direction would be to explore the connection between parametric resurgence and wall-crossing. As is well-known [55,56], the Borel planes of the quantum periods $t(E;\hbar)$ and $t_D(E;\hbar)$ change drastically as we cross certain codimension 1 walls that lie in the complex $E$-plane. From a physics point of view [57,58], these are *walls of marginal stability* and the singularities in the Borel plane correspond to the BPS states of an associated supersymmetric gauge theory. The wall-crossing formula [59] then tells us how the spectrum of BPS states changes as we cross these walls in the moduli space parameterized by $E$. From a resurgence point of view, this phenomenon was also encountered in [60] where these same walls were called *higher order Stokes curves* and the phenomenon was interpreted using the multi-sheeted Borel plane. It would be interesting to revisit this perspective and study the wall-crossing phenomenon also in terms of parametric resurgence and the minimal transseries structures that we uncovered in this work.

# Acknowledgments

We would like to thank Tomás Reis and Marco Serone for useful comments and discussions, and Marcos Mariño for reading and commenting on the final draft of this paper. AvS would like to thank Giulio Bonelli and the TPP group at SISSA for their hospitality during the spring of 2023 when a significant part of this research was conducted.

**Funding information** The research of AvS was supported by the grant OCENW.KLEIN.128, 'A new approach to nonperturbative physics', from the Dutch Research Council (NWO).

# A  On the PNP relation

The PNP relation is essential in establishing the connection between the energy $E$ and the quantum prepotential $F$. In this appendix we provide some more background information on this relation and show its derivation for polynomial potentials. The relation as used in this

---

[27]For $N \geq 4$ it consist of one part, like in our cubic model. For the 'special' $N = 3$ case, it consist of three parts that need to be median resummed.

paper reads

$$c\frac{\partial E}{\partial t} = t\frac{\partial^2 F}{\partial t^2} - \frac{\partial F}{\partial t} + \hbar\frac{\partial^2 F}{\partial t \partial \hbar}, \tag{A.1}$$

or in its integrated version

$$cE = \left(t\frac{\partial}{\partial t} - 2 + \hbar\frac{\partial}{\partial \hbar}\right)F + C(\hbar). \tag{A.2}$$

An alternative common formulation of the PNP relation is one in which the anharmonic coupling $g$ appears. Under the rescaling

$$x \mapsto xg, \qquad p \mapsto pg, \qquad E \mapsto Eg^2, \tag{A.3}$$

we note that the WKB curve $\Sigma$ in (22) remains invariant when we modify the potential appropriately, e.g. we have

$$V_{\text{cubic}} = \frac{x^2}{2} - gx^3, \quad \text{and} \quad V_{\text{dw}} = \frac{x^2}{2}(1 + gx)^2. \tag{A.4}$$

One can then set $\hbar = 1$ and use $g$ as the expansion parameter, thereby turning the energy $E$ and quantum prepotential $F$ into asymptotic expansions in $g^2$ with coefficients depending on $t$. The integrated version of the PNP relation (A.2) then becomes

$$E = \frac{1}{c}\left(\frac{C(g^2)}{g^2} + g^4\frac{\partial F}{\partial g^2}\right), \tag{A.5}$$

where the $t$-independent integration constant now appears as $C(g^2)$.

The PNP relation is essentially a Riemann bilinear identity: in [61] a derivation of this statement was given and a refinement of that same computation was provided in [62]. We would like to improve on these results by repeating the computation for arbitrary genus – which is rather trivial – and by explicitly calculating the constant $c$ which was left undetermined in the former two computations. Before doing so, let us remark that the PNP relation (A.1) can be cast into the form of a *quantum corrected Wronskian condition* [36]:

$$\left(t - \hbar\frac{\partial t}{\partial \hbar}\right)\frac{\partial t_D}{\partial E} - \left(t_D - \hbar\frac{\partial t_D}{\partial \hbar}\right)\frac{\partial t}{\partial E} = c. \tag{A.6}$$

In order to obtain this equation, one exchanges the independent variables $(t, \hbar)$ for $(E, \hbar)$, as explained in [63] section 5. This version of the PNP relation makes the connection to the Riemann bilinear identity manifest, and in what follows we will derive the latter form of the equation, knowing that we can always map it back to (A.1).

For a genus $g$ Riemann surface, the Riemann bilinear identity (see e.g. [64]) for two meromorphic one-forms $\omega_1, \omega_2$ is given by

$$\sum_{i=1}^{g}\left(\oint_{A_i}\omega_1\oint_{B^i}\omega_2 - \oint_{A_i}\omega_2\oint_{B^i}\omega_1\right) = 2\pi i\sum_{\text{poles}}\text{Res}\, f\omega_2, \tag{A.7}$$

where the sum on the right hand side is over poles of both $\omega_1$ and $\omega_2$ and locally near each pole one chooses $f$ such that $\omega_1 = df$. The $A_i, B^j \in H_1(\Sigma)$ form a symplectic basis of one-cycles for the first homology group of the surface: $\langle A_i, A_j\rangle = \langle B^i, B^j\rangle = 0$ and $\langle A_i, B^j\rangle = \delta_i^j$. In our case, the Riemann surface we are interested in is of course the WKB curve. Let the function $S(x, \hbar)$ be a solution to the Riccati equation

$$S^2 - i\hbar\frac{dS}{dx} = 2(E - V(x)). \tag{A.8}$$

Then we can slightly modify the Riccati solution by defining $\tilde{S} = \frac{1}{\hbar}S$ which therefore solves

$$\tilde{S}^2 - i\frac{d\tilde{S}}{dx} = \frac{2}{\hbar^2}(E - V(x)). \tag{A.9}$$

Now choose the one-forms

$$\omega_1 = \left(\frac{\partial \tilde{S}}{\partial \hbar}\right)dx\,,$$
$$\omega_2 = \left(\frac{\partial \tilde{S}}{\partial E}\right)dx\,. \tag{A.10}$$

In what follows, we will show that with these two one-forms, the Riemann bilinear identity (A.7) is equivalent to the quantum corrected Wronskian condition (A.6). In order to compute the residue on the right hand side of the Riemann bilinear identity, we use the argument of [61] which tells us that the Riccati solution only has a pole at infinity.

From the Riccati equation we can deduce the large $x$ scaling behaviour of the one-forms. Without loss of generality, by scaling and shifting $x$ we can write the large $x$ behaviour of a polynomial potential as

$$V(x) \simeq x^d + ax^{d-2} + \mathcal{O}\left(x^{d-3}\right), \tag{A.11}$$

for some number $a$ and an integer $d > 2$. Using dominant balance arguments (see e.g. [65]) one can then deduce the asymptotic behaviour of the two Riccati solutions:

$$\tilde{S} \simeq \pm\frac{i\sqrt{2}}{\hbar}\left(x^{\frac{1}{2}d} + \frac{1}{2}ax^{\frac{1}{2}d-2} + \mathcal{O}\left(x^{\frac{1}{2}d-3}\right)\right). \tag{A.12}$$

The Riccati equation can be differentiated with respect to either $\hbar$ or $E$ to obtain

$$\tilde{S}\frac{\partial \tilde{S}}{\partial \hbar} - \frac{i}{2}\frac{d}{dx}\frac{\partial \tilde{S}}{\partial \hbar} = -\frac{2}{\hbar^3}(E - V(x)),$$
$$\tilde{S}\frac{\partial \tilde{S}}{\partial E} - \frac{i}{2}\frac{d}{dx}\frac{\partial \tilde{S}}{\partial E} = \frac{1}{\hbar^2}. \tag{A.13}$$

Appplying the same dominant balance techiques then yields

$$\tilde{S}\frac{\partial \tilde{S}}{\partial \hbar} \simeq \frac{2}{\hbar^3}V(x),$$
$$\tilde{S}\frac{\partial \tilde{S}}{\partial E} \simeq \frac{1}{\hbar^2}, \tag{A.14}$$

which then gives the scaling behaviour

$$\frac{\partial \tilde{S}}{\partial \hbar} \simeq \mp\frac{i\sqrt{2}}{\hbar^2}x^{\frac{1}{2}d}\left(1 + \frac{1}{2}ax^{-2} + \mathcal{O}(x^{-3})\right),$$
$$\frac{\partial \tilde{S}}{\partial E} \simeq \mp\frac{i}{\hbar\sqrt{2}}x^{-\frac{1}{2}d}\left(1 - \frac{1}{2}ax^{-2} + \mathcal{O}(x^{-3})\right). \tag{A.15}$$

Using these scaling laws, we can compute the right hand side of the Riemann bilinear identity (A.7). Let $\omega_1 = df$ locally for large $x$, then

$$f = \int^x \omega_1 \simeq \mp\frac{i\sqrt{2}}{\hbar}x^{\frac{1}{2}d+1}\left(\frac{1}{\frac{1}{2}d+1} + \frac{ax^{-2}}{d-2} + \mathcal{O}(x^{-3})\right), \tag{A.16}$$

leading to

$$f\omega_2 \simeq \frac{1}{\hbar^3}\left(\frac{2}{d+2}z^{-3} + \frac{4a}{d^2-4}z^{-1} + \mathcal{O}(z^0)\right)dz\,, \tag{A.17}$$

where in the last line we switched to coordinates $z = \frac{1}{x}$. Finally, we take the residue:

$$2\pi i \sum_{\text{poles}} \operatorname{Res} f\, \omega_2 = \frac{16\pi i a}{(d^2 - 4)\hbar^3}\,. \tag{A.18}$$

Note that we count the residue of the pole at infinity ($z = 0$) twice, since $\Sigma$ is a double cover of the Riemann sphere parameterized by $x$ or $z$.

For the left hand side of the Riemann bilinear identity, we note that

$$\oint_\gamma \frac{\partial \tilde{S}}{\partial E} = \frac{1}{\hbar}\frac{\partial \Pi_\gamma}{\partial E}, \quad \text{and} \quad \oint_\gamma \frac{\partial \tilde{S}}{\partial \hbar} = -\frac{1}{\hbar^2}\left(\Pi_\gamma - \hbar\frac{\partial \Pi_\gamma}{\partial \hbar}\right). \tag{A.19}$$

Putting everything together in terms of A-periods $t^i$ and B-periods $t_D^i$, we then obtain

$$\sum_{i=1}^g \left(t^i - \hbar\frac{\partial t^i}{\partial \hbar}\right)\frac{\partial t_D^i}{\partial E} - \left(t_D^i - \hbar\frac{\partial t_D^i}{\partial \hbar}\right)\frac{\partial t^i}{\partial E} = \frac{8a}{d^2 - 4}\,. \tag{A.20}$$

Restricting ourselves to the genus one case that we are interested in in this paper, the index $i$ disappears and we conclude that

$$c = \frac{8a}{d^2 - 4}\,, \tag{A.21}$$

in (A.1) and (A.6). We have checked this quantum corrected Wronskian condition for various potentials of genus one and higher by matching the constant $c$ derived here to the results in [36]. For example, the cubic potential (50) and double well potential (112) can be written in the canonical form (A.11) by an appropriate shift and rescaling, leading to

$$\begin{aligned} V_{\text{cubic}}(x) &= x^3 - \frac{x}{12} - \frac{1}{108}\,, \\ V_{\text{dw}}(x) &= x^4 - \frac{x^2}{2} + \frac{1}{16}\,. \end{aligned} \tag{A.22}$$

This implies that

$$c_{\text{cubic}} = -\frac{2}{15}\,, \quad \text{and} \quad c_{\text{dw}} = -\frac{1}{3}\,, \tag{A.23}$$

which agrees with the PNP relations featured in e.g. [18, 27, 36].

# B   Stokes phenomena for the quantum prepotential from DDP

In the core of this paper we demonstrated two ways in which the alien calculus for the energy transseries could be derived: from mapping the alien calculus of the quantum prepotential (46) to the energy via the PNP relation (section 3.3), or directly from the Delabaere-Dillinger-Pham formula (section 3.4). Here we want to close the circle by showing how the alien calculus of the quantum prepotential can be derived from the Delabaere-Dillinger-Pham formula. We will illustrate this in the cubic potential setting (50) for two Stokes lines: the one running along $\arg(\hbar) = 0$, studied most in the bulk of this paper, and the one along $\arg(\hbar) = \frac{1}{2}\pi$. Let us remark that the reversed argument, deriving the DDP formula from alien calculus of the quantum prepotential, was presented already in [24, 66], both for quantum mechanics and for quantum mirror curves.

**The Stokes phenomenon across arg($\hbar$) = 0:**   Recall that the Delabaere-Dillinger-Pham formula (31) tells us that in the case of the cubic oscillator, when working in the independent variables $(E,\hbar)$, the following Stokes jumps occur along the $\arg(\hbar) = 0$ ray:

$$\begin{aligned}
\mathfrak{S}_0 t(E;\hbar) &= t(E;\hbar) + \frac{\mathrm{i}\hbar}{2\pi} \log\left(1 + \mathrm{e}^{-t_D(E;\hbar)/\hbar}\right), \\
\mathfrak{S}_0 t_D(E;\hbar) &= t_D(E;\hbar).
\end{aligned} \tag{B.1}$$

We see that the dual period is invariant under the action of the automorphism, and so if we apply the pointed alien derivative to it – similar to what we did in section 3.4 – then we obtain

$$0 = \dot{\Delta}_{l\mathcal{A}} t_D(E;\hbar) = \left(\dot{\Delta}_{l\mathcal{A}} F'\right)(t;\hbar)\Big|_{t=t(E;\hbar)} + F''(t(E;\hbar);\hbar)\dot{\Delta}_{l\mathcal{A}} t(E;\hbar), \tag{B.2}$$

where we used (36) to express $t_D(E;\hbar)$ in terms of the quantum prepotential and used the chain rule (18). This identity allows us to express the alien calculus of the (derivative of the) quantum prepotential in terms of the alien calculus of $t(E;\hbar)$. It leads to

$$\left(\dot{\Delta}_{l\mathcal{A}} F'\right)(t;\hbar)\Big|_{t=t(E;\hbar)} = F''(t(E;\hbar));\hbar)\frac{(-1)^{l+1}}{l}\frac{\hbar}{2\pi\mathrm{i}}\mathrm{e}^{-l t_D(E;\hbar)/\hbar}. \tag{B.3}$$

Then we want to integrate both sides with respect to $t$, but $t$ is not an independent variable at this point. Therefore we must first substitute the current independent variable $E$ with $E(t;\hbar)$ and obtain

$$\left(\dot{\Delta}_{l\mathcal{A}} F'\right)(t;\hbar) = F''(t;\hbar)\frac{(-1)^{l+1}}{l}\frac{\hbar}{2\pi\mathrm{i}}\mathrm{e}^{-lF'(t;\hbar)/\hbar}. \tag{B.4}$$

Finally, we can perform the integration with respect to $t$:

$$\left(\dot{\Delta}_{l\mathcal{A}} F\right)(t;\hbar) = \int^t \mathrm{d}t'\,\left(\dot{\Delta}_{l\mathcal{A}} F'\right)(t';\hbar) = \left(-\frac{1}{2\pi\mathrm{i}}\right)F_l^{(l)} + c(\hbar), \tag{B.5}$$

where $F_l^{(l)}$ is defined in (47). Note that the pointed alien derivative $\dot{\Delta}_{l\mathcal{A}}$ maps each formal power series in $\hbar$ to a new formal power series weighted by the transmonomial – schematically:

$$\dot{\Delta}_{l\mathcal{A}} : \mathbb{C}[[\hbar]] \longrightarrow \mathbb{C}[[\hbar]]\,\mathrm{e}^{-l\mathcal{A}/\hbar}. \tag{B.6}$$

Therefore, the integration constant $c(\hbar)$ in (B.5) must be of the form

$$c(\hbar) = \tilde{c}(\hbar)\,\mathrm{e}^{-l\mathcal{A}/\hbar}, \tag{B.7}$$

with $\tilde{c}(\hbar)$ an ordinary perturbative expression in $\hbar$. Note, however, that the $l$-instanton action $l\mathcal{A} = lF_0'(t)$ is a nontrivial function of $t$, which is in contradiction with $c(\hbar)$ being a $t$-independent integration constant. Therefore we conclude that $c(\hbar)$ must vanish, and so

$$\left(\dot{\Delta}_{l\mathcal{A}} F\right)(t;\hbar) = \left(-\frac{1}{2\pi\mathrm{i}}\right)F_l^{(l)}. \tag{B.8}$$

When we substitute $t$ by the classical A-period $t_0(E)$, we obtain the quantum prepotential $F(E;\hbar)$ as defined in (38), which obeys

$$\left(\dot{\Delta}_{l\mathcal{A}} F\right)(E;\hbar) = S_{l\mathcal{A}} F_l^{(l)}. \tag{B.9}$$

This is precisely the action of the pointed alien derivative (46) as proposed in [24], with Stokes constants $S_{l\mathcal{A}} = -\frac{1}{2\pi\mathrm{i}}$.

**The Stokes phenomenon across $\arg(\hbar) = \frac{\pi}{2}$:** Along the positive imaginary axis in the $\hbar$-plane we observe another Stokes phenomenon, which according to the DDP formula (31) can be written as

$$\mathfrak{S}_{\frac{\pi}{2}} t(E;\hbar) = t(E;\hbar),$$
$$\mathfrak{S}_{\frac{\pi}{2}} t_D(E;\hbar) = t_D(E;\hbar) + \hbar \log\left(1 + e^{-2\pi i t(E;\hbar)/\hbar}\right). \tag{B.10}$$

By decomposing the automorphism in terms of pointed alien derivatives with a 'primitive' action $\mathcal{A} = 2\pi i t_0(E)$ which is imaginary, these two equations allow us to deduce the action of those alien derivatives on the dual quantum period:

$$\dot{\Delta}_{l\mathcal{A}} t_D(E;\hbar) = \frac{(-1)^{l+1}}{l} \hbar \, e^{-2\pi i l t(E;\hbar)/\hbar}, \tag{B.11}$$

where we used the fact that $\dot{\Delta}_{l\mathcal{A}}^2 t_D(E;\hbar) = 0$, which follows from (B.10). We then write

$$\dot{\Delta}_{l\mathcal{A}} t_D(E;\hbar) \equiv \dot{\Delta}_{l\mathcal{A}} t_D(t(E;\hbar);\hbar) = \left(\dot{\Delta}_{l\mathcal{A}} F'\right)(t;\hbar)\Big|_{t=t(E;\hbar)}, \tag{B.12}$$

where we used the chain rule, the definition of $t_D(t)$ and the fact that $\dot{\Delta}_{l\mathcal{A}} t(E;\hbar) = 0$, which again follows from (B.10). We then substitute the independent variable $E$ by the quantum mirror map $E(t;\hbar)$ and obtain

$$\dot{\Delta}_{l\mathcal{A}} F'(t;\hbar) = \hbar \frac{(-1)^{l+1}}{l} e^{-2\pi i l t/\hbar}. \tag{B.13}$$

Then we can integrate with respect to $t$ on both sides, and subsequently substitute $t$ with the classical A-period $t_0(E)$ to obtain

$$\dot{\Delta}_{l\mathcal{A}} F(E;\hbar) = \left(-\frac{1}{2\pi i}\right) \frac{(-1)^{l+1}}{l^2} \hbar^2 e^{-2\pi i l t_0(E)/\hbar}. \tag{B.14}$$

When we identify $S_{\mathcal{A}} = -\frac{1}{2\pi i}$ as the Stokes constant, we find a perfect agreement with (46) where $\mathcal{G} = \mathcal{A} = 2\pi i t_0(E)$ and $D = 0$. When we express the Stokes automorphism in terms of alien derivatives, we can now write

$$\mathfrak{S}_{\frac{\pi}{2}} F(E;\hbar) = F(E;\hbar) + \hbar^2 \sum_{n=1}^{\infty} \frac{(-1)^{n+1}}{n^2} \left(-\frac{1}{2\pi i}\right) e^{-2\pi i n t_0(E)/\hbar}. \tag{B.15}$$

The resulting transseries structure on the right hand side matches the one described by (45) for $\mathcal{G} = \mathcal{A} = 2\pi i t_0(E)$ and $D = 0$ with $\tau_n = -\frac{1}{2\pi i}$.

## C   Median resummation

The energy transseries (151) with coefficients $u_{n,m}$ given in table 6 for the double well and in table 8 for the cosine potential were called 'median transseries' for a reason: their median resummation corresponds to the lateral resummations of the $+$ and $-$ resolutions of the energy transseries. Let us elaborate on this statement: first we introduce median resummation (see e.g. [17, 49])

$$\mathcal{S}_{\theta}^{\text{med}} \equiv \mathcal{S}_{\theta^+} \circ \mathfrak{S}_{\theta}^{-1/2} = \mathcal{S}_{\theta^-} \circ \mathfrak{S}_{\theta}^{1/2}, \tag{C.1}$$

where

$$\mathfrak{S}_{\theta}^{\nu} = \exp\left(\nu \sum_{\omega \in \Omega_{\theta}} \dot{\Delta}_{\mathcal{A}_{\omega}}\right). \tag{C.2}$$

The idea of the median resummation operation is to take a well-defined 'average' of the two lateral resummations, thus removing the rather unnatural choice to give $\hbar$ either a positive or a negative imaginary part. For the energy spectrum of the double well (155), the above definition (C.1) then tells us that

$$\mathcal{S}_{0^+}E_\epsilon^{\mathrm{full}}\left(t,+\frac{1}{4\pi\mathrm{i}};\hbar\right)=\mathcal{S}_0^{\mathrm{med}}E_\epsilon^{\mathrm{med}}(t;\hbar)=\mathcal{S}_{0^-}E_\epsilon^{\mathrm{full}}\left(t,-\frac{1}{4\pi\mathrm{i}};\hbar\right), \tag{C.3}$$

and similarly for the cosine potential

$$\mathcal{S}_{0^+}E_\theta^{\mathrm{full}}\left(t,+\frac{1}{2\pi\mathrm{i}};\hbar\right)=\mathcal{S}_0^{\mathrm{med}}E_\theta^{\mathrm{med}}(t;\hbar)=\mathcal{S}_{0^-}E_\theta^{\mathrm{full}}\left(t,-\frac{1}{2\pi\mathrm{i}};\hbar\right). \tag{C.4}$$

In the core of the paper we derived the median transseries by expanding the shift $\Delta t$ in terms of even and odd powers of the instanton transmonomial and subsequently collecting those terms into a resurgent and median part. An alternative way to obtain the median transseries is from the median quantisation condition $D^{\mathrm{med}}=0$ which interpolates between the $D^+=0$ and $D^-=0$ conditions. Let us illustrate this for the cosine potential.[28]

We first change the normalisation of the exact quantisation conditions $D_\theta^\pm=0$ in (160) to

$$\tilde{D}_\theta^\pm=\frac{1}{\sqrt{\mathcal{V}_A^{\mp1}\mathcal{V}_B}}\left(1+\mathcal{V}_A^{\mp1}(1+\mathcal{V}_B)-2\sqrt{\mathcal{V}_A^{\mp1}\mathcal{V}_B}\cos(\theta)\right)=0. \tag{C.5}$$

The reason for this change of normalisaiton is that we now have that

$$\mathfrak{S}_0\tilde{D}_\theta^+=\tilde{D}_\theta^-. \tag{C.6}$$

We can then define the median quantisation condition $D_\theta^{\mathrm{med}}=0$, which is

$$\tilde{D}_\theta^{\mathrm{med}}=\mathfrak{S}_0^{\frac{1}{2}}\tilde{D}_\theta^+=\left(\mathcal{V}_A^{\frac{1}{2}}+\mathcal{V}_A^{-\frac{1}{2}}\right)\sqrt{\frac{1+\mathcal{V}_B}{\mathcal{V}_B}}-2\cos(\theta)=0. \tag{C.7}$$

This is equivalent to

$$\cos\left(\frac{\pi t}{\hbar}\right)=\cos(\theta)\frac{\mathrm{e}^{-\frac{1}{2\hbar}t_D(t;\hbar)}}{\sqrt{1+\mathrm{e}^{-\frac{1}{\hbar}t_D(t;\hbar)}}}. \tag{C.8}$$

The perturbative quantisation condition, for which the entire nonperturbative right hand side can be ignored, then yields the usual

$$t=\hbar\left(N+\frac{1}{2}\right), \tag{C.9}$$

and after introducing the nonperturbative Ansatz $\hat{t}=t+\Delta t$ we find

$$\Delta t=\frac{\hbar}{\pi}\arcsin\left(\cos(\theta)\frac{\mathrm{e}^{-\frac{1}{2\hbar}t_D(t+\Delta t;\hbar)}}{\sqrt{1+\mathrm{e}^{-\frac{1}{\hbar}t_D(t+\Delta t;\hbar)}}}\right), \tag{C.10}$$

which is exactly (170).

---

[28]See also e.g. [67] for the median quantisation conditions of the double and triple well.

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
