# Peer review of "Exact instanton transseries for quantum mechanics"

_SciPost Physics, doi:SciPost Phys. 16, 103 (2024)_

## Round 2 · Referee Report · Anonymous · 2023-12-20

Strengths

1- A clever and complete solution of the energy trans-series for three one-dimensional quantum mechanical models.
2- Very clear presentation of the results.
3- A nice and quick review of the resurgence theory and the known results of the resurgent structure of quantum prepotential, based on which the results of this paper are derived.

Weaknesses

None.

Report

In this paper, the authors investigate non-perturbative corrections to the energy spectra of one-dimensional quantum mechanical systems for positive real values of the Planck constant ℏ, and derive the full trans-series solution of the energy spectra including instanton corrections to all orders. The authors first solved this problem for the cubic model, and they did this by solving the energy E from the non-perturbative quantisation conditions using the Lagrange inversion theorem. The energy trans-series is, as usually, ambiguous when resummed via Borel resummation, and the ambiguity is cured by Stokes transformation across the Stokes ray along the positive real axis of the ℏ plane. They then clarify the resurgent structure of the energy trans-series by extracting from the trans-series the Stokes constants for Borel singularities along the positive real axis. These Stokes constants are also computed in two other approaches, by deriving them from known Stokes constants of quantum prepotentials which are related to the energy spectra by the PNP relation, and by calculating them from the known Stokes transformations of the quantum periods. All these results agree with each other.

The authors then extend their studies to two other quantum mechanical models, the symmetric double well model and the cosine potential model. In both examples, the authors are able to write down the full trans-series solution to the energy spectra, and to extract the complete set of Stokes constants for Borel singularities along the positive real axis. One important difference from the example of the cubic model is that in the cubic model, all the non-perturbative corrections in the trans-series are related to, or resurge from, the perturbative series via Stokes transformations; in other words, the trans-series is mininal. In the symmetric double well model and the cosine potential model, it is found not to be the case. The full energy trans-series is rather a sum of infinitely many minimal trans-series. But surprisingly, the authors find that the full energy trans-series can factorise to a product of a minimal trans-series, and another trans-series known as the median trans-series.

Although the quantisation conditions for the three models considered here have been known for some time, it is for the first time that the full energy trans-series have been written down. In addition, although it was clear that the ambiguity of the Borel resummation energy trans-series was cured by Stokes transformation, it was only demonstrated for the first few instanton sectors, and it is generalised in this paper to all instanton sectors by working out the complete set of Stokes constants along the positive real axis. Finally, the finding that the full energy trans-series of the double well model and the cosine potential model factorise is quite revealing.

Given the originality of the findings in this paper, I recommend the publication of this paper in this journal. It would be better though some explicit examples for (3.43) could be provided, presumably by repeating the calculation in (3.39).

Requested changes

1- Some explicit examples of (3.43), presumably by repeating the calculation in (3.39).

  • validity: top
  • significance: high
  • originality: high
  • clarity: top
  • formatting: perfect
  • grammar: perfect

Author:  Alexander van Spaendonck  on 2024-02-07  [id 4301]

(in reply to Report 1 on 2023-12-20)

Dear referee, thank you for reading our paper and for your positive review. Regarding your requested change: for deriving (3.43) we did not exploit the PNP relation as we did in (3.39), but rather used the results of (3.41) in combination with the fact that the pointed alien derivatives are operators satisfying the Leibniz rule. We have supplemented the text around equation (3.43) with some extra sentences to explain this more carefully and added the suggestion that the same result can also be derived via the PNP relation, as you rightly suggest.

---

## Round 2 · Referee Report · Anonymous · 2024-1-25

Strengths

1. Detailed pedagogical treatment of the resurgent transseries structure of the energy in an interesting class of genus 1 quantum mechanical models.
2. Nice summary and illustrative examples of the power of resurgent alien calculus as an efficient way to characterise all perturbative and nonperturbative features of such quantum mechanical models.
3. Clear illustrative examples of the relation to quantum prepotentials and the associated perturbative-nonperturbative (PNP) relation.

Report

This is a very good paper which I recommend for publication in SciPost. The authors have analysed thoroughly the transseries structure arising for the energy in a special class of simple genus 1 quantum mechanical models. The paper is quite pedagogical in spirit, but it also contains some interesting new technical results. The authors give a detailed analysis of 3 different model potentials: the cubic potential, the symmetric double-well potential, and the periodic cosine (Mathieu) potential. In the first model they identify a very compact general structure of the energy transseries when expressed in terms of the coupling (hbar) and the full quantum action (t). They call this a "minimal transseries": see equation (3.45). A particularly nice part of this analysis is the clear distinction the authors draw between the "exact quantisation condition", familiar from exact WKB, and the perturbative-nonperturbative PNP relation which is ultimately a more geometric result.
In the further two model systems the authors show that the minimal transseries structure is generalized via extra contributions which they refer to as the "median transseries". This is clearly explained and illustrated by these two models, in the former case in terms of the action of parity and in the latter case in terms of the Bloch phase. Mathematically, this highlights the way the transseries structure is influenced by both the Stokes phenomenon in hbar and also the monodromy in the action.
Taken together, these three models illustrate comprehensively the rich structure of transseries arising in quantum mechanical spectral problems. Appendix A presents a good summary of the geometric basis of the PNP relation for the simple class of genus 1 polynomial potentials, and hints at how this generalises to more complicated genus 1 models, and to higher genus. Appendix B summarises how the quantum prepotential relates to the classic work of Delabaere, Dillinger and Pham, and Appendix C gives more details on the definition of the median transseries appearing in the double well and periodic potential problems. These appendices provide useful background information for readers interested in more technical detail.

Requested changes

This is an optional comment which I feel could strengthen the paper if the authors are able to clarify: I found the notation in the introduction of the perturbative quantum prepotential between equations (2.37) and (2.44) to be a little cryptic, which may be a barrier for readers who are unfamiliar with the field. Perhaps the authors could consider finding a way to clarify this discussion, or even move it to later in the paper after an example has been done.

  • validity: -
  • significance: -
  • originality: -
  • clarity: -
  • formatting: -
  • grammar: -

Author:  Alexander van Spaendonck  on 2024-02-07  [id 4302]

(in reply to Report 2 on 2024-01-25)

Dear referee, thank you for taking the time to read our paper and your positive words. Regarding your suggestion: we agree that our introduction of the transseries solution to the quantum prepotential between equations (2.37) and (2.44) is somewhat short and glosses over some subtle details from the Gu-Marino paper where this transseries was derived. For our purposes, the main results that we are after are equations (2.45) and (2.46), and thus in order to accommodate your suggestions we have clarified this now in the text. In particular, we have rephrased the text to emphasise that this is merely a short summary and added a footnote stating : "The key result that we are after here is action of the alien derivatives on the perturbative quantum prepotential $F(E;\hbar)$ shown in (2.45). For the purpose of understanding the main results of this paper, the finer details of the quantum prepotential transeries $\hat F$ presented here can be skipped upon first reading."

---

## Round 3 · Referee Report · Anonymous · 2024-2-7

Report

The author has addressed my request, and clarified the derivation of (3.43). So I recommend the publication of this paper.

---

## Editorial Decision

published